# UNPAIRED PANORAMIC IMAGE-TO-IMAGE TRANSLATION LEVERAGING PINHOLE IMAGES

## ABSTRACT

In this paper, we tackle the challenging task of unpaired panoramic Image-to-Image translation (Pano-I2I) for the first time. This task aims to learn a mapping between unpaired panoramic source and non-panoramic target domains to modify naive 360° panoramic images. However, it is difficult to use existing I2I methods due to two main challenges. Firstly, panoramas inherit geometric distortions, which pose challenges for methods based on a narrow field-of-view. Secondly, accessing panoramic datasets encompassing various weather conditions or times for training purposes is severely limited. To address these challenges, we propose a novel I2I model tailored for mitigating panoramic distortion that harnesses readily obtainable pinhole images as the target domain for training. We introduce a versatile encoder and distortion-free discrimination that efficiently bridges the large domain gap between panoramic and pinhole images by simultaneously encoding them in a consolidated structure. It allows our model to learn style mappings while overcoming significant geometric differences between the source and target domains. Moreover, we carefully design spherical position embedding, sphere-based rotation augmentation, and its ensemble to address the discontinuities at the panorama edges. Comprehensive experiments verify that our framework effectively translates panoramic street views from daytime to night, rainy, and twilight scenes by referring to the holistic style of pinhole data. Our method also shows superior results in both maintaining structural coherence and rotation equivariance, clearly surpassing the existing I2I methods in qualitative and quantitative results.

## 1 INTRODUCTION

Image-to-image translation (I2I) aims to modify an input image aligning with the style of the target domain, preserving the original content from the source domain. This paradigm enables numerous applications, such as colorization, style transfer, domain adaptation, data augmentation, *etc* (Huang et al., 2018a; Mariani et al., 2018; Murez et al., 2018; Huang & Belongie, 2017; Isola et al., 2017). However, existing I2I has been used to synthesize pinhole images with narrow field-of-view (FoV), which limits the scope of applications considering diverse image-capturing devices.

Panoramic 360° cameras have recently grown in popularity, which enables many applications, *e.g.*, AR/VR, autonomous driving, and city map modeling (Micusik & Kosecka, 2009; Anderson et al., 2018; Caruso et al., 2015; Yogamani et al., 2019). Unlike pinhole images of narrow FoV, panoramic images (briefly, panoramas) capture the entire surroundings, providing richer information with 360°×180° FoV. Translating panoramas into other styles can enable novel applications, such as immersive view generations or enriching user experiences with robust car-surrounding recognition (de La Garanderie et al., 2018; Yang et al., 2020; 2021b; Ma et al., 2021).

However, naively applying conventional I2I methods for pinhole images (Park et al., 2020a; Shen et al., 2019; Choi et al., 2018; 2020; Zheng et al., 2021; Jeong et al., 2021; Kim et al., 2022) to panoramas can significantly distort the geometric properties of panoramas. As existing I2I methods assume the source and target domains only differ in style conditions, they have limitations in distinguishing between style and inherent distortion where the two domains have additional geometric gaps. One may project the panoramas into pinhole images to apply the conventional methods. However, it costs considerable computation since sparse projections (Chou et al., 2018; Zhu et al., 2020) cannot cover the whole scene due to the narrow FoV of pinhole images. In addition, the

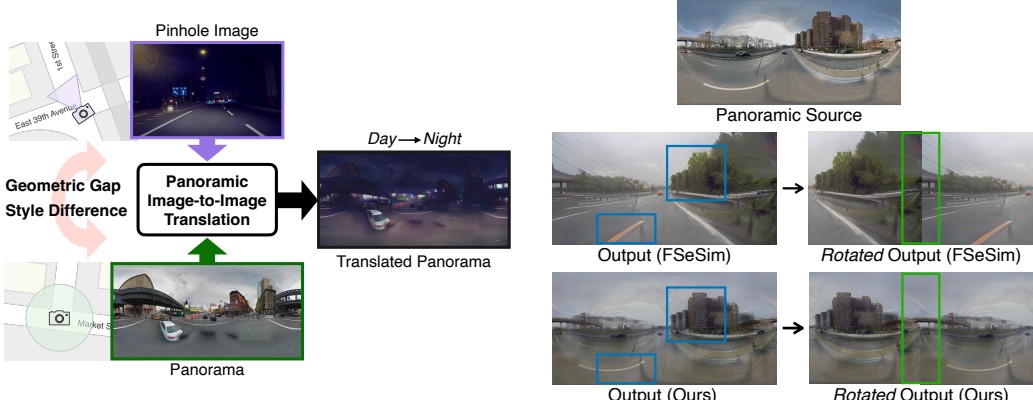

Figure 1: **Illustration of our problem formulation.** Our Pano-I2I is trained on panoramas in the daytime as the source domain and pinhole images with diverse conditions as the target domain, where the two domains have significant geometric and style gaps.

Figure 2: **Failure cases of an existing method, FSeSim.** The pinhole-like structure (blue) and structural-, style-discontinuity at edges (green) are spotted. In contrast, ours achieves a rotation-equivariant and panoramic structure. We visualize rotated outputs ($\theta=180°$) to highlight discontinuity.

discontinuity problem at edges (left-right boundaries in panorama) requires panorama-specific modeling, as in the other tasks, *e.g.*, panorama depth estimation, panoramic segmentation, and panorama synthesis (Shen et al., 2022; Zhang et al., 2022b; Chen et al., 2022).

Another challenge of panoramic image translation is the absence of sufficient panorama datasets covering diverse conditions. Especially for I2I, panoramas obtained under diverse conditions such as sunny, rainy, and night are needed to consider various target styles. However, compared to the pinhole images, panoramas are captured by a specially-designed camera (360° camera) or post-processed using multi-view images obtained from the calibrated cameras. Note that panoramic images for the Street View services are usually taken during the day (Mirowski et al., 2018). Instead of constructing a costly new panorama dataset, it would be desirable to leverage existing pinhole image datasets with various conditions as style guidance, whose goal and results are illustrated in Fig. 1. We present typical failure cases of existing approaches, FSesim (Zheng et al., 2021) in Fig. 2. The result from FSeSim shows the collapsed structure having a pinhole-like structure (blue), as it simply overfits the geometric nature of the pinhole target. These indicate naively applying existing I2I methods has limited capacity to deal with the large domain gap between panoramas and pinhole images, resulting in disentangling content and style failures. Additionally, it has structural- and style-discontinuity at edges (green). Based on the above analysis, we seek to expand the applicability of I2I to panoramic images by employing existing pinhole image datasets as style domain, dubbed the *Unpaired Panoramic Image Translation Leveraging Pinhole Images*, shortly, Pano-I2I.

To address geometric deformation in panoramas, we adopt deformable convolutions (Zhu et al., 2019) to our encoders, with different offsets for panoramas and pinhole images to reflect the geometric differences between the source and target. We propose a distortion-free discrimination that attenuates the geometric differences to handle the large gap between the source and target domain. In addition, we adopt panoramic rotation augmentation and ensemble techniques to solve discontinuity problems at edges, considering that a 360° panorama should be continuous at boundaries.

We validate the proposed approach by evaluating the panorama dataset, StreetLearn (Mirowski et al., 2018), and the pinhole datasets (Shen et al., 2019; Sakaridis et al., 2019). Our proposed method significantly outperforms all existing methods across various target conditions from the pinhole datasets. We also provide ablation studies to analyze the components. Our main contributions are:

- For the first time, to the best of our knowledge, we propose the panoramic I2I task and approach translating panoramas with pinhole images as a target domain.

- We present distortion-free discrimination along with our versatile encoder, spherical positional embedding, and sphere-based rotation augmentation to deal with a large geometric gap between the source and the target and discontinuity at the edges of panoramas.

- Pano-I2I notably outperforms the previous methods in the quantitative evaluations for style relevance and structural similarity, providing qualitative analyses.

## 2 RELATED WORK

**Image-to-image translation.** Different from early works requiring paired dataset (Isola et al., 2017), the seminal works (Zhu et al., 2017) enabled unpaired source/target training (*i.e.*, learning without the grount-truth pair). Some works enable multimodal learning (Huang et al., 2018b; Lee et al., 2018a; 2020), multi-domain learning (Choi et al., 2018; 2020; Wu et al., 2019), translating across domains with large discrepancies (Amodio & Krishnaswamy, 2019; Yang et al., 2022) for simple datasets, and instance-aware learning (Bhattacharjee et al., 2020; Jeong et al., 2021; Kim et al., 2022) in complex scenes. Nevertheless, existing I2I methods are restrictive to specific source-target pairs; Previous methods (Lee et al., 2018a; Huang et al., 2018b) have translated images by reintegrating disentangled content and style of the scene to avoid geometric deformation but focused solely on the object figuration as the content. Namely, they are limited to handling 1) geometric variations (*e.g.*, part deformation, viewpoint, scale) between source and target domains. Our approach introduces a robust framework to an unpaired setting, even with geometric differences in complex scenes, and 2) rotational equivalence for panoramas. On the other hand, several works have adopted the architecture of vision transformers (Dosovitskiy et al., 2021) to image generation (Lee et al., 2021; Jiang et al., 2021b; Zhang et al., 2022a). Being capable of learning long-range interactions, the transformer is often employed for high-resolution image generation (Esser et al., 2021; Zhang et al., 2022a), or complex scene generation (Wang et al., 2021; Kim et al., 2022). For instance, InstaFormer (Kim et al., 2022) proposed to use transformer for I2I, capturing global consensus in complex street-view scenes. Recently, diffusion models have been employed in image editing (Ye et al., 2023; Zhang & Agrawala, 2023; Brooks et al., 2023; Mou et al., 2023) thanks to their generative capacity. However, those methods are still limited to considering wide geometric discrepancies between source (panoramas) and target (pinhole) domains and rotation equivariance of panoramas.

**Panoramic image modeling.** Panoramic images from 360° cameras provide a thorough view of the scene with a wide FoV, beneficial in understanding the scene holistically. Since panorama itself faces challenges (FoV, focal length, aberration), this brought out unique architectures or modeling techniques, *e.g.,* depth estimation (Tateno et al., 2018; Jin et al., 2020; Pintore et al., 2021; Peng & Zhang, 2023), perception (Yang et al., 2021a; Orhan & Bastanlar, 2022; Zheng et al., 2023), aerial-to-panorama generation (Lu et al., 2020; Wu et al., 2022). A common practice to address distortions in panoramas is to project an image into other formats of 360° images (*e.g.,* equirectangular, cubemap) (Cheng et al., 2018; Wang et al., 2018; Yang et al., 2019), and some works even combine both equirectangular and cubemap projections with improving performance (Jiang et al., 2021a; Wang et al., 2020). However, they do not consider the properties of 360° images, such as the connection between the edges of the images and the geometric distortion caused by the projection. Several works leverage narrow FoV projected images (Lee et al., 2018b; Yang et al., 2018; de La Garanderie et al., 2018), but they require many projected images (*e.g.*, 81 images (Lee et al., 2018b)), which is an additional burden. To deal with such discontinuity and distortion, recent works introduce modeling in spherical domain (Esteves et al., 2018; Cohen et al., 2018), projecting an image to local tangent patches with minimal geometric error. It is proved that leveraging transformer architecture in 360° image modeling reduces distortions caused by projection and rotation (Cho et al., 2022). For this reason, recent approaches (Ranftl et al., 2021; Rey-Area et al., 2022; Yun et al., 2022; Shen et al., 2022; Chen et al., 2022) used the transformer achieving global structural consistency.

## 3 METHODOLOGY

### 3.1 PROBLEM DEFINITION

We use the panoramic domain as a source representing content structures and the pinhole domain as a target style. Formally, given a panorama of the source domain $\mathcal{X}$, Pano-I2I learns a mapping function that translates its style into target pinhole domain $\mathcal{Y}$ retaining the content and structure of the panorama. Unlike the previous I2I methods (Zhu et al., 2017; Park et al., 2020a;b; Jiang et al., 2020; Choi et al., 2020; Liu et al., 2021; Zheng et al., 2021; Jeong et al., 2021; Kim et al., 2022) that have source and target domains both in a narrow FoV of pinhole images, our setting varies both in style and structure; the source domain as panoramas with wide FoV captured in the daytime, and the target domain as pinhole images in diverse conditions with narrow FoV. Here, existing I2I methods designed fail to preserve the panoramic structure of the content, since their feature disentanglements cannot effectively separate the style from target content when both *geometric* and *style* differences

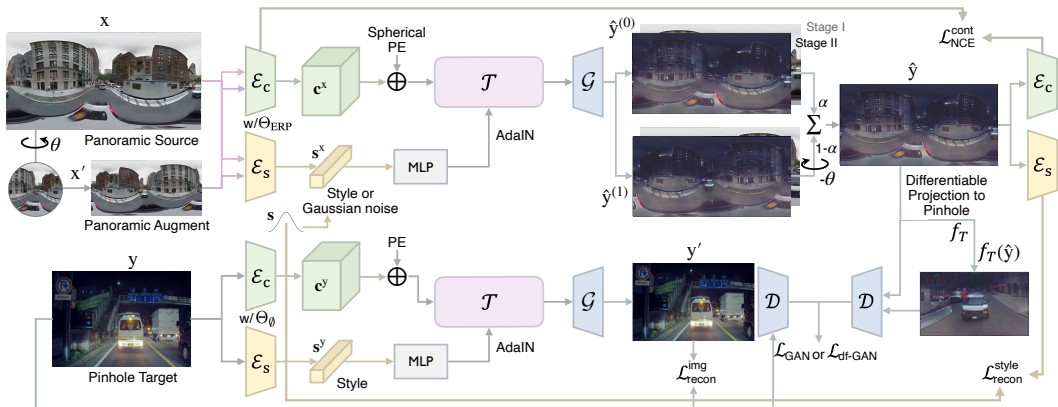

Figure 3: **Overall network configuration of Pano-I2I**. Given panoramas as the source domain, we disentangle the content and translate its style aligned to the target domain. For training, pinhole datasets are used as targets referring to styles. Panoramic augmentation and ensemble are also introduced to preserve the spherical structure of panoramas. In our framework, Stage I only learns to reconstruct the panorama source, and Stage II learns panorama translation.

exist. We empirically observed that 1) the outputs of the I2I methods are pinhole-like, as shown in Fig. 2, and 2) pinhole image-based network design that does not consider the spherical structure of 360° causes discontinuity at the left-right boundaries and low visual fidelity.

## 3.2 ARCHITECTURE DESIGN

**Overall architecture.** On a high level, the proposed method consists of a shared content encoder $\mathcal{E}_c$, a shared style encoder $\mathcal{E}_s$ to estimate the disentangled representation, a transformer encoder $\mathcal{T}$ to mix the style and content through AdaIN (Huang & Belongie, 2017) layers, and a shared generator $\mathcal{G}$ and discriminator $\mathcal{D}$ to generate the translated image. Our transformer encoder block consists of a multi-head self-attention layer and a feed-forward MLP with GELU (Vaswani et al., 2017).

In specific, to translate an image $\mathbf{x}$ in source domain $\mathcal{X}$ to target domain $\mathcal{Y}$, we first extract a content feature map $\mathbf{c}^{\mathbf{x}} \in \mathbb{R}^{h \times w \times l_c}$ by the content encoder from $\mathbf{x}$, with height $h$, width $w$, and $l_c$ content channels, and receive a random latent code $\mathbf{s} \in \mathbb{R}^{1 \times 1 \times l_s}$ from Gaussian distribution $\mathcal{N}(0, \mathbf{I}) \in \mathbb{R}^{l_s}$, which is used to control the style of the output with the affine parameters for AdaIN (Huang & Belongie, 2017) layers in transformer encoders. Finally, we get the output image by $\hat{\mathbf{y}} = \mathcal{G}(\mathcal{T}(\mathbf{c}^{\mathbf{x}}, \mathbf{s}))$. In the following, we will explain our key ingredients – panoramic modeling in the content encoder, style encoder and the transformer, distortion-free discrimination, and sphere-based rotation augmentation and its ensemble – in detail.

**Panoramic modeling in encoders.** In Pano-I2I, latent spaces are shared to embed domain-invariant content or style features as done in (Lee et al., 2018a). Namely, the content and style encoders, $\mathcal{E}_c$ and $\mathcal{E}_s$, respectively, take either panoramas or pinhole images as inputs. However, the geometric distortion gap from the different FoVs prevents the encoders from understanding each corresponding structural information. To overcome this, motivated from (Shen et al., 2022; Yun et al., 2022), we use the specially-designed deformable convolution layer (Zhu et al., 2019) at the beginning of the content and style encoders by adjusting the offset reflecting the characteristics of the image types. Specifically, given a panorama, the deformable convolution layer can be applied directly to the panorama by deriving an equirectangular plane (ERP) offset (Yun et al., 2022), $\Theta_{\mathrm{ERP}}$, that considers the panoramic geometry. To this end, we project a tangential local patch $\mathcal{P}$ of a 3D-Cartesian domain into ERP to obtain the corresponding ERP offset, $\Theta_{\mathrm{ERP}}$, as follows:

$$\Theta_{\mathrm{ERP}}(\theta, \phi) = f_{\mathrm{SPH \to ERP}}\Big( f_{\mathrm{3D \to SPH}}\big( \mathcal{P} \times R(\theta, \phi) / ||\mathcal{P} \times R(\theta, \phi)||_2 \big) \Big), \tag{1}$$

where $R(\theta, \phi)$ indicates the rotation matrix with the longitude $\theta \in [0, 2\pi]$ and latitude $\phi \in [0, \pi]$, $f_{\mathrm{SPH \to ERP}}$ indicates the conversion function from spherical domain to ERP domain, and $f_{\mathrm{3D \to SPH}}$ indicates the conversion function from 3D-Cartesian domain to spherical domain, as in PAVER (Yun et al., 2022). To be detailed, we first project the tangential local patch $\mathcal{P}$ to the corresponding

spherical patch on a unit sphere $S^2$, aligning the patch center to $(\theta, \phi) \in S^2$. Notice that the number of $\mathcal{P}$ is $H \times W$ with the stride of 1 and proper paddings, while the center of $\mathcal{P}$ corresponds to the kernel center. Then, we obtain the relative spherical coordinates from the center point and all discrete locations in $\mathcal{P}$. Finally, these positions are projected to the ERP domain as offset points. We compute such 2D-offset points $\Theta_{\text{ERP}} \in \mathbb{R}^{2 \times H \times W \times \ker_h \times \ker_w}$ for each kernel location, and fixed to use them throughout training and test phase.

Unlike basic convolution, which has a square receptive field with limited capability to deal with geometric distortions in panoramas, our deformable convolution with fixed offsets can encode panoramic structure. Note that we repurpose the deformable convolution as an adapter to our shared encoders to handle the panorama source and pinhole target simultaneously by different offsets, while PAVER adopts this to patch embedding for the panoramic transformer. This approach is efficient as the same encoders learn the compact representation that explains both types of images. By considering the context of the source and target within the shared network, our approach ensures that the output images reflect the desired style in a manner that is consistent with the source and target. For pinhole image encoding, $\Theta_{\text{ERP}}$ is replaced to zero-offset $\Theta_{\varnothing}$ in both content and style encoders, which are vanilla convolutions. Please refer to the details in Appendix A.2.

**Panoramic modeling in the translator.** After extracting the content features from the source image, we first patchify the content features to be processed through transformer blocks, then add positional embedding (PE) (Vaswani et al., 2017; Rahaman et al., 2019), and process to generator. We represent the center coordinates of the previous patchified grids as $(i_{\text{p}}, j_{\text{p}})$ corresponding to the $p$-th patch. As two kinds of inputs $\{\mathbf{x}, \mathbf{y}\}$ —panorama, pinhole image, for each— have different structural properties, we adopt sinusoidal PE in two ways: using 2D PE and spherical PE (SPE), respectively.

To start with, we define the fixed PE in transformer (Vaswani et al., 2017) as $\gamma(\cdot)$, a sinusoidal mapping into $\mathbb{R}^{2K}$ as follows:

$$\gamma(a) = \{(\sin(2^{k-1}\pi a), \cos(2^{k-1}\pi a))\}, \quad k \in \{1, 2, ..., K\} \tag{2}$$

for an input scalar $a$. Based on this, we define the 2D PE for common pinhole images as follows:

$$\text{PE} = \text{concat}(\gamma(i_{\text{p}}), \gamma(j_{\text{p}})). \tag{3}$$

Following the previous work (Chen et al., 2022), we consider the $360°$ spherical structure of panorama presenting a spherical positional embedding for the center position $(i_{\text{p}}, j_{\text{p}})$ of each grid defined as follows, further added to the patch embedded tokens to work as explicit guidance:

$$\text{SPE} = \text{concat}(\gamma(\theta), \gamma(\phi)), \quad \text{where } \theta = (2i_{\text{p}}/h - 1)\pi, \ \phi = (2j_{\text{p}}/w - 1)\pi/2, \tag{4}$$

where each patch has width $w$ and height $h$. Since SPE explicitly provides cyclic spatial guidance and the relative spatial relationship between the tokens, it helps to maintain rotational equivariance for the panorama by encouraging structural continuity at boundaries in an $360°$ input. On the other hand, the previous spherical methods used the standard PE (Yun et al., 2022; Zhang et al., 2022b) or limited to employ the SPE for implicit guidance (Chen et al., 2022) as conditioning in text-to-image generation, which does not provide token-wise spatial information. This positional embedding is added to patch-embedded tokens and further processed into transformer encoders with AdaIN.

**Distortion-free discrimination.** As our source and target domains exhibit two distinct features in geometric structure and style, directly applying the existing I2I method for panoramic I2I guided by pinhole images brings severe structural collapse with artifacts, affecting the synthesizing quality. We speculate that the structural collapse stems from breaking the balance of discriminators. Concretely, while the loss of discriminator in I2I typically learns to distinguish real data $\mathbf{y}$ and fake data $\hat{\mathbf{y}}$, mainly focusing on style difference, in our task, there is an additional large deformation gap and FoV difference between $\mathbf{y}$ and $\hat{\mathbf{y}}$ that confuses what to discriminate, as illustrated in Fig. 1.

To address this issue, we present a distortion-free discrimination technique. The key idea is to transform a randomly selected region of a panorama into a pinhole image while maintaining the degree of FoV (see right-bottom section in Fig. 3). Specifically, we propose to adopt a panorama-to-pinhole image conversion $f_T$ by a rectilinear projection (Lee et al., 2018b). To obtain a narrow FoV (pinhole-like) image with $f_T$, we first select a viewpoint in the form of longitude and latitude coordinates $(\theta, \phi)$ in the spherical coordinate system, where $\theta$ and $\phi$ are randomly selected from $[0, 2\pi]$ and $[0, \pi]$, respectively, and extract a narrow FoV region from the $360°$ panorama image by a differentiable projection function $f_T$. To further improve the discriminative ability of our model, we

adopt a weighted sum of the original discrimination and the proposed discrimination to encourage the model to learn more robust features by considering both the original full-panoramas and pinhole-like converted panoramas (refer to Eqn. 7).

**Sphere-based rotation augmentation and ensemble.** We introduce a panoramic rotation-based augmentation since a different panorama view consistently preserves the content structure without the discontinuity problem at the left-right boundaries. Given a panorama $\mathbf{x}$, a rotated image $\mathbf{x}'$ is generated by horizontally rotating angle $\theta$. We efficiently implement this rotation by rolling the images in the ERP space without the burden of ERP→SPH→ERP projections since both are *effectively* the same operation. The rotation angle is randomly sampled in $(0, 2\pi)$, where the step size is $2\pi/10$. Such rotation is also reflected in SPE by adding the rotation angle $\theta$ to help the model learn the horizontal cyclicity of panoramas. Later, the translated images $\hat{\mathbf{y}}^{(0)}$ and $\hat{\mathbf{y}}^{(1)}$ from the generator with $\mathbf{x}$ and $\mathbf{x}'$, respectively, are blended together, after rotating back with $-\theta$ for $\hat{\mathbf{y}}^{(1)}$ of course, to generate the final ensemble output $\hat{\mathbf{y}}$:

$$\hat{\mathbf{y}} = (\hat{\mathbf{y}}^{(0)} + \hat{\mathbf{y}}^{(1)'})/2 \tag{5}$$

where $\hat{\mathbf{y}}^{(1)'}$ is indicates $-\theta$ rotated version of $\hat{\mathbf{y}}^{(1)}$. Thus, the result $\hat{\mathbf{y}}$ has a smoother boundary than the results predicted alone, mitigating discontinuous edge effects.

## 3.3 LOSS FUNCTIONS

**Adversarial loss.** We minimize the distribution discrepancy between two different features (Good-fellow et al., 2014; Mirza & Osindero, 2014) by adversarial loss. We adopt this to learn the translated image $\hat{\mathbf{y}} = \mathcal{G}(\mathcal{T}(\mathcal{E}_c(\mathbf{x}, \Theta_{\text{ERP}}), \mathbf{s}))$ and the image $\mathbf{x}$ from $\mathcal{X}$ to have indistinguishable distribution to preserve panoramic contents, defined as:

$$\mathcal{L}_{\text{GAN}} = \mathbb{E}_{\mathbf{x} \sim \mathcal{X}}[\log(1 - \mathcal{D}(\hat{\mathbf{y}}))] + \mathbb{E}_{\mathbf{y} \sim \mathcal{Y}}[\log \mathcal{D}(\mathbf{y})], \tag{6}$$

with the R1 regularization (Mescheder et al., 2018) to enhance training stability. To consider the panoramic distortion-free discrimination using the panorama-to-pinhole conversion $f_T$, we define additional adversarial loss as follows:

$$\mathcal{L}_{\text{df-GAN}} = \mathbb{E}_{\mathbf{x} \sim \mathcal{X}}[\log(1 - \mathcal{D}(f_T(\hat{\mathbf{y}})))] + \mathbb{E}_{\mathbf{y} \sim \mathcal{Y}}[\log \mathcal{D}(\mathbf{y})]. \tag{7}$$

**Content loss.** To maintain the content between the source image $\mathbf{x}$ and translated image $\hat{\mathbf{y}}$, we exploit the spatially-correlative loss (Zheng et al., 2021) to define a content loss, with an augmented source $\mathbf{x}_{\text{aug}}$. To get $\mathbf{x}_{\text{aug}}$, we apply structure-preserving transformations to $\mathbf{x}$. This helps preserve the structure and learn the spatially-correlative map (Zheng et al., 2021) based on patchwise infoNCE loss (Oord et al., 2018), since it captures the domain-invariant structure representation. Denoting that $\hat{\mathbf{v}}$ as spatially-correlative map of the query patch from $\mathbf{c}^{\hat{\mathbf{y}}} = \mathcal{E}_c(\hat{\mathbf{y}}, \Theta_{\text{ERP}})$, we pick the pseudo-positive patch sample $\overset{+}{\mathbf{v}}$ from $\mathbf{c}^{\mathbf{x}} = \mathcal{E}_c(\mathbf{x}, \Theta_{\text{ERP}})$ in the same position of the query patch $\hat{\mathbf{v}}$, and the negative patches $\overline{\mathbf{v}}$ from the other positions of $\mathbf{c}^{\mathbf{x}}_{\text{aug}}$ and $\mathbf{c}^{\mathbf{x}}$, except for the position of query patches $\hat{\mathbf{v}}$. We first define a score function $\ell(\cdot)$ at the $l$-th convolution layer in $\mathcal{E}_c$:

$$\ell(\hat{\mathbf{v}}, \overset{+}{\mathbf{v}}, \overline{\mathbf{v}}) = -\log\left[\frac{\exp(\hat{\mathbf{v}} \cdot \overset{+}{\mathbf{v}} / \tau)}{\exp(\hat{\mathbf{v}} \cdot \overset{+}{\mathbf{v}} / \tau) + \sum_{s \in S \backslash s} \exp(\hat{\mathbf{v}} \cdot \overline{\mathbf{v}}^s / \tau)}\right], \tag{8}$$

where $\tau$ is a temperature parameter. Then, the overall content loss function is defined as follows:

$$\mathcal{L}_{\text{NCE}}^{\text{cont}} = \mathbb{E}_{\mathbf{x} \sim \mathcal{X}} \sum_l \sum_s \ell(\hat{\mathbf{v}}_l^s, \overset{+}{\mathbf{v}}_l^s, \overline{\mathbf{v}}_l^{S \backslash s}), \tag{9}$$

where the index $s \in \{1, 2, ..., S_l\}$ and $S$ is a set of patches in each $l$-th layer, and $S \backslash s$ indicates the indices except $s$.

**Image reconstruction loss.** We enhance the disentanglement between content and style in a manner that our $\mathcal{G}$ can reconstruct an image for domain $\mathcal{Y}$. To be specific, $\mathbf{y}$ is fed into content encoder $\mathcal{E}_c$ and style encoder $\mathcal{E}_s$ to obtain a content feature map $\mathbf{c}^{\mathbf{y}} = \mathcal{E}_c(\mathbf{y}, \Theta_\varnothing)$ and a style code $\mathbf{s}^{\mathbf{y}} = \mathcal{E}_s(\mathbf{y}, \Theta_\varnothing)$. We then compare the reconstructed image $\mathcal{G}(\mathcal{T}(\mathbf{c}^{\mathbf{y}}, \mathbf{s}^{\mathbf{y}}))$ with $\mathbf{y}$ as follows:

$$\mathcal{L}_{\text{recon}}^{\text{img}} = \mathbb{E}_{\mathbf{y} \sim \mathcal{Y}}[\|\mathcal{G}(\mathcal{T}(\mathbf{c}^{\mathbf{y}}, \mathbf{s}^{\mathbf{y}})) - \mathbf{y}\|_1]. \tag{10}$$

**Style reconstruction loss.** In order to better learn disentangled representation, we compute L1 loss between the style code from the translated image and input panorama,

$$\mathcal{L}_{\text{ref-recon}}^{\text{sty}} = \mathbb{E}_{\mathbf{x} \sim \mathcal{X}}[\|\mathcal{E}_{\text{s}}(\hat{\mathbf{y}}, \Theta_{\text{ERP}}) - \mathcal{E}_{\text{s}}(\mathbf{x}, \Theta_{\text{ERP}})\|_1]. \tag{11}$$

We also define the style reconstruction loss to reconstruct the style code $\mathbf{s}$, used to generate $\hat{\mathbf{y}}$. Note that the style code $\mathbf{s}$ is randomly sampled from Gaussian distribution, not extracted from an image.

$$\mathcal{L}_{\text{rand-recon}}^{\text{sty}} = \mathbb{E}_{\mathbf{x} \sim \mathcal{X}, \mathbf{y} \sim \mathcal{Y}}[\|\mathcal{E}_{\text{s}}(\hat{\mathbf{y}}, \Theta_{\text{ERP}}) - \mathbf{s}\|_1]. \tag{12}$$

### 3.4 TRAINING STRATEGY

**Stage I: Panorama reconstruction.** Since there is no publicly available large-scale outdoor panorama data under various weather or season conditions, we cannot use panoramas as a style reference. In addition, in order to share the same embedding space in content and style, the network must be able to process pinhole images and panoramas simultaneously.

For the stable training of Pano-I2I, the training procedure is split into two stages corresponding with different objectives. In Stage I, we pretrain the content and style encoders $\mathcal{E}_{\text{c,s}}$, transformer $\mathcal{T}$, generator $\mathcal{G}$, and discriminator $\mathcal{D}$ using the panorama dataset only. Given a panorama, the parameters of our network are optimized to reconstruct the original with adversarial and content losses, and style reconstruction loss. As the network learns to reconstruct the input self again, we use the style feature represented by a style encoder instead of the random style code. For $\mathcal{L}_{\text{GAN}}$ in Stage I, the original discriminator receives $\mathbf{x}$ instead of $\mathbf{y}$ as an input. The total objective in Stage I as follows:

$$\mathcal{L}_{\text{StageI}} = \mathcal{L}_{\text{GAN}} + \lambda_{\text{cont}} \mathcal{L}_{\text{NCE}}^{\text{cont}} + \lambda_{\text{sty}} \mathcal{L}_{\text{ref-recon}}^{\text{sty}}, \tag{13}$$

where $\lambda_{\{*\}}$ denotes balancing hyperparameters that control the importance of each loss.

**Stage II: Panoramic I2I guided by pinhole image.** In Stage II, the whole network is fully trained with robust initialization by Stage I. Compared to Stage I, panorama and pinhole datasets are all used in this stage. Concretely, the main difference is that; (1) original discrimination is combined with our distortion-free discrimination as a weighted sum, (2) the style code is sampled from the Gaussian distribution to translate the panorama, (3) the panoramic rotation-based augmentation and its ensemble technique are leveraged to enhance the generation quality. Therefore, the total objective in Stage II is defined as:

$$\mathcal{L}_{\text{Stage2}} = \lambda_{\text{df-GAN}} \mathcal{L}_{\text{df-GAN}} + (1 - \lambda_{\text{df-GAN}}) \mathcal{L}_{\text{GAN}} + \lambda_{\text{cont}} \mathcal{L}_{\text{NCE}}^{\text{cont}} + \lambda_{\text{sty}} \mathcal{L}_{\text{rand-recon}}^{\text{sty}} + \lambda_{\text{recon}} \mathcal{L}_{\text{recon}}^{\text{img}}.$$

Notice that all coefficients for the losses are set to 1, except for $\lambda_{\text{NCE}}^{\text{cont}}$, which is set as 15, and $\lambda_{\text{df-GAN}}$ is set as 0.8 for Stage II. Those coefficients were commonly used in all training datasets. Please refer to Appendix A.3 for other training details.

## 4 EXPERIMENTS

### 4.1 EXPERIMENTAL SETUP

**Datasets.** We conduct experiments on the panorama dataset, StreetLearn (Mirowski et al., 2018), as the source domain, and a standard street-view dataset for I2I, INIT (Shen et al., 2019) and Dark Zurich (Sakaridis et al., 2019), as the target domain. StreetLearn provides 360° outdoor 56k panoramas taken from the Google Street View. Although INIT consists of four conditions (sunny, night, rainy, and cloudy), we use two conditions, night and rainy, since the condition of the StreetLearn is captured during the daytime, including sunny and cloudy. We use the *Batch1* of INIT dataset, a total of 62k images for the four conditions. Dark Zurich has three conditions (daytime, night, and twilight), a total of 8779 images, and we use night and twilight.

**Metrics.** For quantitative comparison, we report the Fréchet Inception Distance (FID) metric (Heusel et al., 2017) to evaluate style relevance, and the structural similarity (SSIM) index (Wang et al., 2004) metric to evaluate the panoramic content preserving. Considering that the structure of outputs tends to become pinhole-like in panoramic I2I tasks, we measure the FID metric after applying panorama-to-pinhole projection ($f_T$) for randomly chosen horizontal angle $\theta$ and fixed vertical

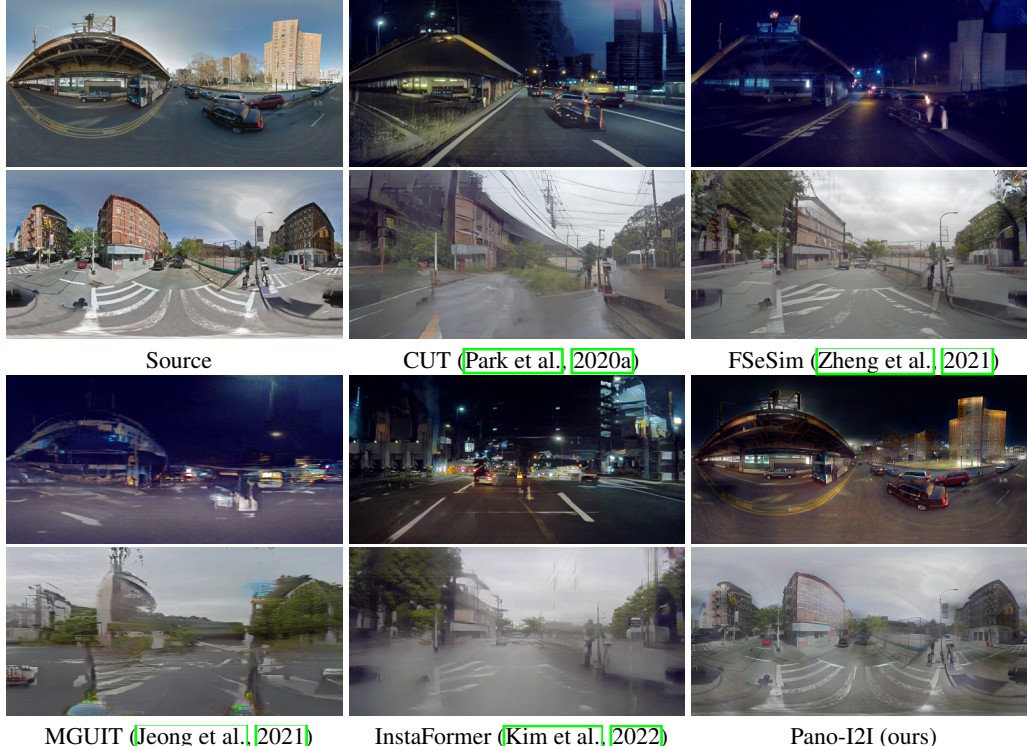

Source       CUT (Park et al., 2020a)       FSeSim (Zheng et al., 2021)

MGUIT (Jeong et al., 2021)     InstaFormer (Kim et al., 2022)     Pano-I2I (ours)

Figure 4: **Qualitative comparison** on StreetLearn dataset (day) to INIT dataset (night, rainy): (top to bottom) day→night, and day→rainy results. Among the methods, Pano-I2I (ours) preserves object details well and shows realistic results.

Table 1: **Quantitative evaluation** on the translated panoramas from StreetLearn to INIT.

| Methods | Day→Night | | Day→Rainy | |
|---|---|---|---|---|
| | FID↓ | SSIM↑ | FID↓ | SSIM↑ |
| CUT (Park et al., 2020a) | 131.3 | 0.232 | 119.8 | 0.439 |
| FSeSim (Zheng et al., 2021) | 106.0 | 0.309 | 110.3 | 0.541 |
| MGUIT (Jeong et al., 2021) | 129.9 | 0.156 | 141.5 | 0.268 |
| InstaFormer (Kim et al., 2022) | 151.1 | 0.201 | 136.2 | 0.495 |
| Pano-I2I (ours) | **94.3** | **0.417** | **86.6** | **0.708** |

Table 2: **Quantitative evaluation** on the translated panoramas from StreetLearn to Dark Zurich.

| Methods | Day→Night | | Day→Twilight | |
|---|---|---|---|---|
| | FID↓ | SSIM↑ | FID↓ | SSIM↑ |
| CUT (Park et al., 2020a) | 152.3 | 0.277 | 196.7 | 0.289 |
| FSeSim (Zheng et al., 2021) | 133.8 | 0.315 | 138.8 | 0.420 |
| MGUIT (Jeong et al., 2021) | 187.3 | 0.156 | 191.5 | 0.124 |
| InstaFormer (Kim et al., 2022) | 165.6 | 0.213 | 141.7 | 0.296 |
| Pano-I2I (ours) | **120.2** | **0.431** | **126.6** | **0.520** |

angle $\phi$ as 0 with a fixed FoV of 90°, for consistent viewpoint with the target images. Notice that the SSIM shows the degree of content preservation because it measures the structural similarity between the original panorama and the translated panorama based on luminance, contrast, and structure.

**Comparison methods.** We compare our approach against the state-of-the-art I2I methods, including MGUIT (Jeong et al., 2021) and InstaFormer (Kim et al., 2022), CUT (Park et al., 2020a), and FS-eSim (Zheng et al., 2021). We train them with same datasets as ours. Since MGUIT and InstaFormer require bounding box annotations to train their models, we exploit pretrained YOLOv5 (Jocher et al., 2021) model to generate pseudo bounding box annotations.

## 4.2 IMPLEMENTATION DETAIL

We formulate the proposed method with vision transformers (Dosovitskiy et al., 2021) inspired by InstaFormer (Kim et al., 2022), but without instance-level approaches due to the absence of ground-truth bounding box annotations. We design our content and style encoders, a transformer encoder, the generator-and-discriminator for our GAN losses based on (Kim et al., 2022), where all modules are learned from scratch. Details on architecture details are presented in Appendix A.

### 4.3 EXPERIMENTAL RESULT

**Qualitative evaluation.** In Fig. 4, we qualitatively compare our method with other I2I methods. We observe all the other methods (Park et al., 2020a; Jiang et al., 2020; Zheng et al., 2021; Jeong et al., 2021; Kim et al., 2022) fail to synthesize reasonable panoramic results and show obvious inconsistent output regarding either structure or style in an image. Moreover, previous methods recognize structural discrepancies between source and target domains as style differences, indicating failed translation results that change like pinhole images. Surprisingly, in the case of 'day→night', all existing methods fail to preserve the objectness as a car or building. We conjecture that they cannot efficiently deal with the large domain gap in 'day→night,' thus naively learning to follow the target distribution without considering the context from the source. By comparison, our method shows the overall best performance in visual quality, preserving panoramic content, and structural- and style-consistency. In particular, we observe the ability of our discrimination design to learn distortion-tolerated features. The qualitative results on the Dark Zurich dataset are in Appendix E.2, and comparisons with recent I2I methods and diffusion-based methods are in Appendix E.3.

**Quantitative evaluation.** Tab. 1 and Tab. 2 show the quantitative comparison in terms of FID (Heusel et al., 2017) and SSIM (Wang et al., 2004) index metrics. Our method consistently outperforms the competitive methods in all metrics, demonstrating that Pano-I2I successfully captures the style of the target domain while preserving the panoramic contents. Notably, our approach exhibits significant improvements in terms of SSIM. In contrast, previous methods perform poorly in terms of SSIM compared to our results, which is also evident from the qualitative results presented in Fig. 4.

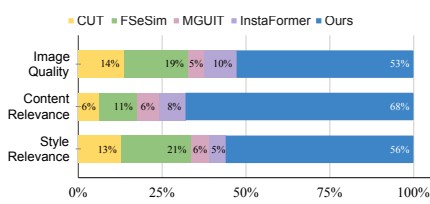

Figure 5: **User study results.**

**User study.** We conducted a user study to compare the subjective quality. We randomly select 10 images for each task (day→night, rainy) on INIT dataset, and let 60 users sort all the methods regarding "overall image quality", "content preservation from the source", and "style relevance with the target, considering the context from the source". As seen in Fig. 5, our method has a clear advantage on every task. More details are in Appendix F.

### 4.4 ABLATION STUDY

In Tab. 3, we show quantitative results for the ablation study on the day→night task on the INIT dataset. In particular, we analyze the effectiveness of our 1) distortion-free discrimination, 2) ensemble technique, 3) two-stage learning scheme, and 4) spherical positional embedding (SPE) and deformable convolution. We observe all the techniques and components contribute to performance improvement in terms of style relevance and content preservation, and the impact of distortion-free discrimination is substantially effective in handling geometric deformation. We validate our design choices by qualitative results of the ablation study with more detailed discussions in Appendix C.

Table 3: **Evaluation on ablation study.**

| ID | Methods | FID↓ | SSIM↑ |
|----|---------|------|-------|
| (I) | Pano-I2I (ours) | **94.3** | **0.417** |
| (II) | (I) - Distortion-free $\mathcal{D}$ | 105.6 | 0.321 |
| (III) | (I) - Ensemble technique | 96.8 | 0.390 |
| (IV) | (I) - Two-stage learning | 120.8 | 0.376 |
| (V) | (I) - SPE, deform conv | 94.5 | 0.355 |

## 5 CONCLUSION

In this paper, we introduce an experimental protocol and a dedicated model for the panoramic Image-to-Image Translation that considers 1) the structural properties of the panoramic images and 2) the lack of outdoor panoramic scene datasets. To this end, we design our model to take panoramas as the source, and pinhole images with diverse conditions as the target, raising the large geometric variance between the source and target domains as a major challenging point. To mitigate these issues, we propose distortion-aware panoramic modeling techniques and distortion-free discriminators to stabilize adversarial learning. Additionally, exploiting the cyclic property of panoramas, we propose to rotate and fuse the synthesized panoramas, resulting in the panorama output with a continuous view. We demonstrate the success of our method in translating realistic images in several benchmarks and look forward to future works that use our proposed experimental paradigm for panoramic image translation with non-pinhole camera inputs using diverse sets of pinhole image datasets.

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
