APPENDIX

In this document, we describe implementation details, additional quantitative results, more qualitative results of the ablation study, the pseudo-code of our two-stage learning, additional experimental results, the details of the user study, and limitations for our "Unpaired Panoramic Image-to-Image Translation Leveraging Pinhole Images."

## A  IMPLEMENTATION DETAILS

### A.1  ARCHITECTURE DETAILS

We summarize the detailed network architecture of our method in Tab. 4. We borrow the architecture of content encoder, style encoder, transformer blocks, and generator from InstaFormer(Kim et al., 2022) and discriminator from StarGANv2 (Choi et al., 2020). 'Layers for style encoder' inside the Encoder table indicates the end of the content encoder, while the style encoder has the same structure as the content encoder except for additional adaptive average pooling (AdaptiveAvgPool) and Conv-4, as shown below. Also, unlike the content encoder, the style encoder does not contain any normalization layer. DeformConv indicates Deformable convolution with offset, and (.) in the convolution indicates the zero-padding.

Table 4: **Network architecture of Pano-I2I.**

**Encoder**

| Layer | Parameters ($\mathtt{in}, \mathtt{out}, \mathtt{k}, \mathtt{s}, \mathtt{p}$) | Output shape ($C \times H \times W$) |
|---|---|---|
| DeformConv-1 (Reflection) | $(3, 64, 7, 1, 3)$ | $(64, 256, 512)$ |
| InstanceNorm | - | $(64, 256, 512)$ |
| ReLU | - | $(64, 256, 512)$ |
| Conv-2 (Zeros) | $(64, 128, 3, 1, 1)$ | $(128, 256, 512)$ |
| InstanceNorm | - | $(128, 256, 512)$ |
| ReLU | - | $(128, 256, 512)$ |
| Downsample | - | $(128, 128, 256)$ |
| Conv-3 (Zeros) | $(128, 256, 3, 1, 1)$ | $(256, 128, 256)$ |
| InstanceNorm | - | $(256, 128, 256)$ |
| ReLU | - | $(256, 128, 256)$ |
| DownSample | - | $(256, 64, 128)$ |
| **Layers for style encoder** | | |
| AdaptiveAvgPool | - | $(256, 1, 1)$ |
| Conv-4 | $(256, 8, 1, 1, 0)$ | $(8, 1, 1)$ |

**Transformer Encoder**

| Layer | Parameters ($\mathtt{in}, \mathtt{out}$) | Output shape ($C$) |
|---|---|---|
| AdaptiveInstanceNorm | - | $(1024)$ |
| Linear-1 | $(1024, 3072)$ | $(3072)$ |
| Attention | - | $(1024)$ |
| Linear-2 | $(1024, 1024)$ | $(1024)$ |
| AdaptiveInstanceNorm | - | $(1024)$ |
| Linear-3 | $(1024, 4096)$ | $(4096)$ |
| GELU | - | $(4096)$ |
| Linear-4 | $(4096, 1024)$ | $(1024)$ |

**Generator**

| Layer | Parameters ($\mathtt{in}, \mathtt{out}, \mathtt{k}, \mathtt{s}, \mathtt{p}$) | Output shape ($C \times H \times W$) |
|---|---|---|
| UpSample | - | $(256, 128, 256)$ |
| Conv-1 (Zeros) | $(256, 128, 3, 1, 1)$ | $(128, 128, 256)$ |
| LayerNorm | - | $(128, 128, 256)$ |
| ReLU | - | $(128, 128, 256)$ |
| UpSample | - | $(128, 256, 512)$ |
| Conv-2 (Zeros) | $(128, 64, 3, 1, 1)$ | $(64, 256, 512)$ |
| LayerNorm | - | $(64, 256, 512)$ |
| ReLU | - | $(64, 256, 512)$ |
| Conv-3 (ReflectionPad) | $(64, 3, 7, 1, 3)$ | $(3, 256, 512)$ |
| Tanh | - | $(3, 256, 512)$ |

## A.2 Deformable convolution

Our deformable convolution finds the fixed offset for ERP format, as in PAVER (Yun et al., 2022) and PanoFormer (Shen et al., 2022). As mentioned in the main paper, the deformable convolution layer is applied on the equirectangular (ERP) format of panorama image by deriving an ERP plane offset $\Theta_{\text{ERP}} \in \mathbb{R}^{2 \times H \times W \times \ker_h \times \ker_w}$ for each kernel location (here, the kernel size is 7×7) that considers the panoramic geometry. After obtaining the offset for once, we keep the kernel shape on the tangent fixed. The conversion function from 3D-Cartesian domain to spherical domain and spherical domain to ERP domain:

$$f_{\text{3D} \rightarrow \text{SPH}}(x, y, z) = (\arctan \frac{y}{x}, \arctan \frac{\sqrt{x^2 + y^2}}{z}), f_{\text{SPH} \rightarrow \text{ERP}}(\theta, \phi) = (\frac{W}{2\pi}\phi, \frac{H}{\pi}\theta), \quad (14)$$

where W, H is width and height for the panoramic input, respectively, and $\theta \in [0, 2\pi], \phi \in [0, \pi]$.

The rotation matrix is as follows:

$$R(\theta, \phi) = \begin{pmatrix} \cos\phi\cos\theta & -\cos\phi\sin\theta & \sin\theta \\ \sin\theta & \cos\theta & 0 \\ \sin\phi\cos\theta & -\sin\phi\sin\theta & \cos\phi \end{pmatrix}. \quad (15)$$

## A.3 Training details

We employ the Adam optimizer, where $\beta_1 = 0.9$ and $\beta_2 = 0.99$, for 100 epochs using a step decay learning rate scheduler. We also set a batch size of 8 for Stage I, and 4 for Stage II for each GPU. The initial learning rate is 1e-4. All coefficients for the losses are set to 1, except for $\lambda_{\text{NCE}}^{\text{cont}}$, which is set as 15, and $\lambda_{\text{df}-\text{GAN}}$ is set as 0.8 for Stage II. The number of negative patches for content loss is 255. The training images are resized to 256×512. The initial learning rate is 1e-4, and the model is trained on 8 Tesla V100 with batch size 8 for Stage I and 4 for Stage II. The trained weights and code will be made publicly available.

## A.4 Notation

We provide the notations that are used in the main paper, in Tab. 5.

Table 5: **Our notations are summarized.**

| Symbol | Definition |
|---|---|
| $\mathbf{x}$ | Content image from source domain (panorama) |
| $\mathbf{y}$ | Style image from target domain (pinhole image) |
| $\hat{\mathbf{y}}$ | Translated image (panorama) |
| $\mathcal{E}_c$ | Content encoder with Deformable Conv |
| $\mathcal{E}_s$ | Style encoder with Deformable Conv |
| $\mathcal{T}$ | Transformer encoder |
| $\mathcal{G}$ | Generator |
| $\mathcal{D}$ | Discriminator |
| $\Theta$ | Offset used in deformable layer |
| $\theta$ | rotation angle in augmentation |
| $S$ | Length of one side of tangential square patches |
| $W, H$ | Sizes of 360° image input (*e.g.,*, $W = 512, H = 256$) |
| $w, h$ | Number of patches along width and height |
| $l$ | Number of channels per feature |

A.5 EVALUATION DETAILS

**Fréchet Inception Distance (FID)** (Heusel et al., 2017) is computed by measuring the mean and variance distances between the generated and real images in the Inception feature space. We used the default setting of FID measurement provided in [1]. In the main paper, we sampled 10 times for 1000 test images. Therefore, we computed the FID for each sampled set and averaged the scores to get the final result, and evaluated FID between target images and output images to measure style relevance. In addition, considering that the structure of outputs tends to become pinhole-like in panoramic I2I tasks, we measure the FID metric after applying panorama-to-pinhole projection ($f_T$) for randomly chosen horizontal angle $\theta$ and fixed vertical angle $\phi$ as 0 with a fixed FoV of 90° to the output images, for consistent viewpoint with the target images. We visualize some examples of the projected images, compared with other methods in Fig. 6. It should be noted that we measure FID between the original target images and the projected output images.

However, we noticed that the synthesized images of existing methods seem to follow not only the style of target images but also the pinhole structure and its contents (*e.g.*, appearance of road, buildings, cars). In this regard, the higher FID score for style relevance does not guarantee better stylization results in this task. Therefore, we additionally adopt the FID metric to measure structural distributions between source images and output images. To exclude the style information, we conduct such measurement in grayscale image format, shown in Tab. 6. We indicate such FID measurement as $\text{FID}_c$.

**Structural Similarity Index Measure (SSIM)** (Wang et al., 2004) is a widely used full-reference image quality assessment (IQA) measure, which measures the similarity between two images, where one of them is the reference image. We adopt the SSIM metric to measure the structural similarity between the source image and the output image.

**Pinhole Image Datasets** In our framework, we exploit pinhole images as the target domain, while we use panoramas as the source domain. We show examples of pinhole image datasets: INIT and Dark Zurich, in Fig. 7.

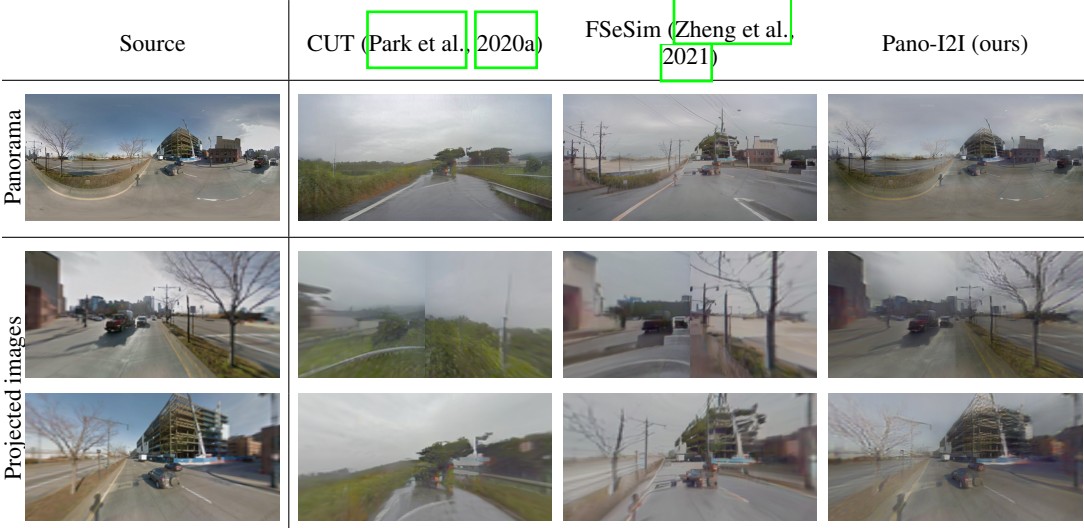

Figure 6: **Visualization of panorama-to-pinhole image conversion.** We visualize the input and the output images (first row) and the examples of projected images (second row and third row) where $\phi$ is fixed as 0. Note that the projected source images are not used as input to the networks.

---

[1]https://github.com/mseitzer/pytorch-fid

| INIT (night) | INIT (rainy) | Dark Zurich (night) | Dark Zurich (twilight) |

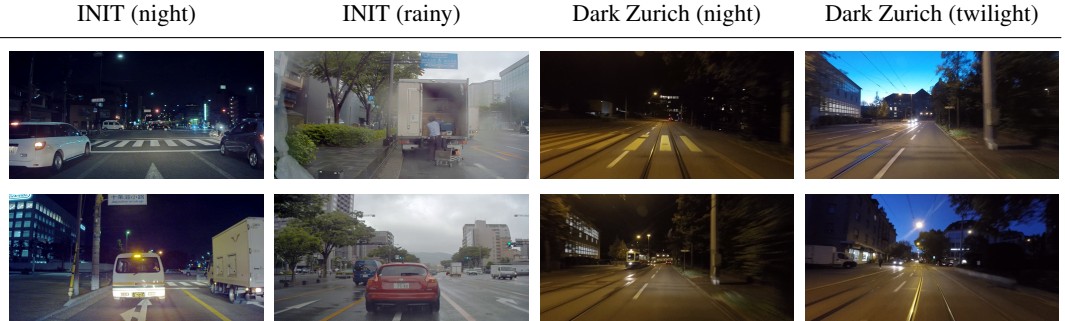

Figure 7: **Examples of pinhole image datasets.** We visualize the samples of INIT dataset and Dark Zurich dataset, which are used as target domains.

## B  MORE QUANTITATIVE RESULTS

In this section, we report additional quantitative results in Tab. 6 to complement the main results. As explained in section A.5, we additionally adopt $FID_c$ metric to measure structural distributions between source panoramas and output panoramas, unlike in the main paper. In order to exclude the style information, we transform images into a grayscale format.

Table 6: **Quantitative evaluation** in terms of the matching of feature distributions (FID (Heusel et al., 2017) metric) on StreetLearn dataset (Mirowski et al., 2018) to INIT dataset (Shen et al., 2019). Note that the purpose of FID measurement is different from the main paper; here we additionally adopt FID metric to measure structural distance.

| Method | Day→Night | Day→Rainy |
|---|---|---|
| | $FID_c \downarrow$ | $FID_c \downarrow$ |
| CUT (Park et al., 2020a) | 225.60 | 153.72 |
| FSeSim (Zheng et al., 2021) | 179.28 | 136.44 |
| MGUIT (Jeong et al., 2021) | 433.17 | 147.38 |
| InstaFormer (Kim et al., 2022) | 231.38 | 149.91 |
| Pano-I2I (ours) | **85.13** | **85.49** |

## C  ADDITIONAL RESULTS ON ABLATION STUDY

In the main paper, we have examined the impacts of distortion-free discrimination, rotational ensemble, SPE and deformable convolution, and two-stage learning with quantitative and qualitative results. In this section, we provide additional visual results on day→night (StreetLearn (Mirowski et al., 2018) to INIT (Shen et al., 2019)) with more comprehensive discussions.

As seen in Fig. 8, our full model can maintain the boundary with the high-quality generation, successfully preserving the panoramic structure. It should be noted that the visual results are $\theta=180°$ rotated to highlight the continuity at the edges. Especially, we can observe the ability of our discrimination design to generate distortion-tolerate outputs, and the results without ensemble technique fail to represent consistent style within an image. As also seen in SSIM in Tab. 3 in the main paper, our distortion-free $\mathcal{D}$, deformable conv, and SPE mainly address the geometric- and distributon-gap between source and target, and our $\mathcal{D}$ is the most important for preserving content.

SPE and deformable convolution also help preserve structural continuity. The power of ensemble for style continuity at the boundaries of an image is well described in Fig. 8. The results without SPE and deformable convolution show a limited capability to capture structural continuity. Two-stage learning is crucial for preserving style relevance as it helps disentanglement, observed by FID. Additionally, the worst SSIM value is better than any of other methods (in Tab. 3). This demonstrates all of each contribution boost the performance, which is not trivial.

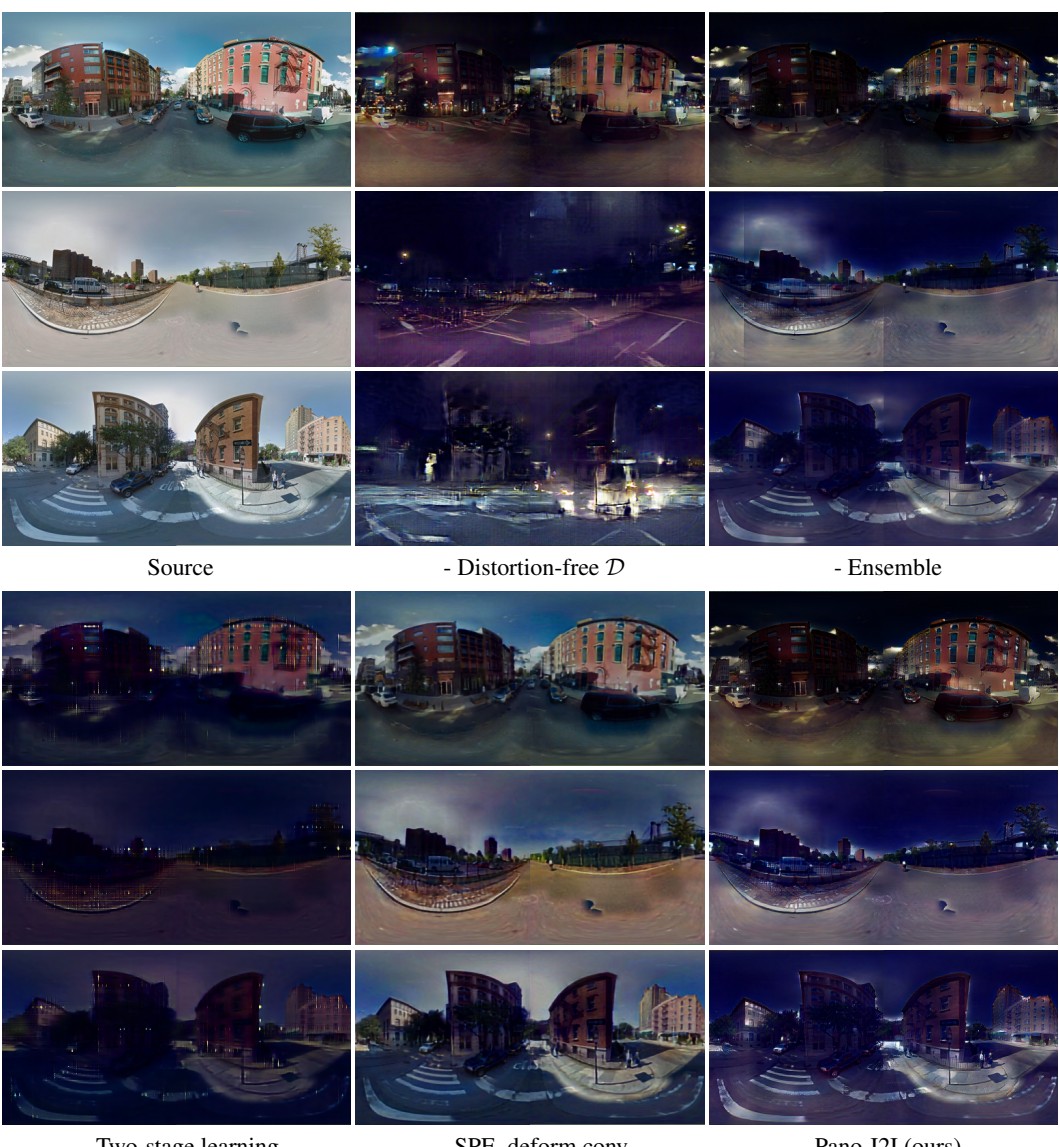

Source        - Distortion-free $\mathcal{D}$        - Ensemble

- Two-stage learning        - SPE, deform conv        Pano-I2I (ours)

Figure 8: **Additional qualitative evaluation on ablation study.**

# D    TRAINING ALGORITHMS

Here we provide the training algorithms for our stage-wise learning in pseudo code forms and diagrams, where the first stage aims to reconstruct input panoramas, and the second stage learns to translate panoramas to have the style of pinhole images, while preserving panoramic structure.

## D.1    TRAINING ALGORITHM FOR STAGE-I

---

**Algorithm 1:** Pseudo code for stage-I

---

**Inputs :**

$\mathcal{X}$      Source domain (panorama)

$\mathcal{E}_c, \mathcal{E}_s$      Content and style encoders with a deformable Convolution layer

$\mathcal{T}$      Transformer encoder

$\mathcal{G}$      Generator

$\mathcal{D}$      Discriminator

$\xi$      Initial parameters for $\mathcal{E}_c, \mathcal{E}_s, \mathcal{T}, \mathcal{G}$, and $\mathcal{D}$

$\Theta_{\mathrm{ERP}}$      Equirectangular plane offset used in the deformable conv layer

$\theta$      Rotation angle in panoramic augmentation

$S_l$      Number of patches in each $l$-th layer

$N$      Total number of optimization steps

$\eta_n$      Learning rate for $n$-th optimization step

$\alpha$      Panoramic ensemble ratio

$\gamma(\cdot)$      Sinusoidal mapping s.t. $\gamma(a) = \{(\sin(2^{k-1}\pi a), \cos(2^{k-1}\pi a))|k = 1, ..., K\}$

$\ell(\cdot)$      InfoNCE loss with a positive pair $(\mathbf{v}, \mathbf{v}^+)$, negative pairs $(\mathbf{v}, \mathbf{v}^-)$, and a temperature $\tau$ s.t.

$$\ell(\hat{\mathbf{v}}, \mathbf{v}^+, \mathbf{v}^-) = -\log\left[\frac{\exp(\mathbf{v}\cdot\mathbf{v}^+/\tau)}{\exp(\mathbf{v}\cdot\mathbf{v}^+/\tau)+\sum\exp(\mathbf{v}\cdot\mathbf{v}^-/\tau)}\right]$$

**for** $n = 1$ **to** $N$ **do**

    $\mathcal{B} \leftarrow \{\mathrm{x}|\mathrm{x} \sim \mathcal{X}\}$

    **for** $(\mathrm{x}) \in \mathcal{B}$ **do**

        $\mathrm{x}' \leftarrow \mathrm{PanoramicAugment}(\mathrm{x}, \theta)$

        $\mathbf{c}^{\mathrm{x}}, \mathbf{c}^{\mathrm{x}'} \leftarrow \mathcal{E}_c(\mathrm{x}, \Theta_{\mathrm{ERP}}), \mathcal{E}_c(\mathrm{x}', \Theta_{\mathrm{ERP}})$

        $\mathbf{s}^{\mathrm{x}} \leftarrow \mathcal{E}_s(\mathrm{x}, \Theta_{\mathrm{ERP}})$

        $\hat{\mathbf{y}}^{(0)}, \hat{\mathbf{y}}^{(1)} \leftarrow \mathcal{G}(\mathcal{T}(\mathbf{c}^{\mathrm{x}}, \mathbf{s}^{\mathrm{x}})), \mathcal{G}(\mathcal{T}(\mathbf{c}^{\mathrm{x}'}, \mathbf{s}^{\mathrm{x}}))$

        $\hat{\mathbf{y}} \leftarrow \alpha \cdot \hat{\mathbf{y}}^{(0)} + (1 - \alpha) \cdot \mathrm{PanoramicAugment}(\hat{\mathbf{y}}^{(1)}, -\theta)$

        $\mathbf{c}^{\mathrm{x}}, \mathbf{c}^{\hat{y}} \leftarrow \mathcal{E}_c(\mathrm{x}, \Theta_{\mathrm{ERP}}), \mathcal{E}_c(\hat{y}, \Theta_{\mathrm{ERP}})$

        $\hat{\mathbf{v}}, \mathbf{v}^+, \mathbf{v}^- \leftarrow \mathrm{PatchSample}_{\mathrm{pos}}(\mathbf{c}^{\hat{y}}), \mathrm{PatchSample}_{\mathrm{pos}}(\mathbf{c}^{\mathrm{x}}), \mathrm{PatchSample}_{\mathrm{neg}}(\mathbf{c}^{\mathrm{x}})$

        $\mathbf{s}^{\hat{y}} \leftarrow \mathcal{E}_s(\hat{y}, \Theta_{\mathrm{ERP}})$

        $\mathcal{L}_{\mathrm{GAN}} \leftarrow \log(1 - \mathcal{D}(\hat{\mathbf{y}})) + \log \mathcal{D}(\mathbf{x})$

        $\mathcal{L}_{\mathrm{NCE}}^{\mathrm{cont}} \leftarrow \sum_l \sum_s \ell(\hat{\mathbf{v}}, \mathbf{v}^+, \mathbf{v}^-)$

        $\mathcal{L}_{\mathrm{ref\text{-}recon}}^{\mathrm{style}} \leftarrow \|\mathbf{s}^{\hat{y}} - \mathbf{s}^{\mathrm{x}}\|_1$

    **end**

    $\delta\xi \leftarrow \frac{1}{N}\sum_{(\mathrm{x,y})\in\mathcal{B}} \partial_\xi \mathcal{L}_{\mathrm{GAN}} + \lambda_{\mathrm{cont}}\mathcal{L}_{\mathrm{NCE}}^{\mathrm{cont}} + \lambda_{\mathrm{style}}\mathcal{L}_{\mathrm{ref\text{-}recon}}^{\mathrm{style}}$

    $\xi \leftarrow \mathrm{optimizer}(\xi, \delta\xi, \eta_n)$

**end**

---

## D.2 TRAINING ALGORITHM FOR STAGE-II

---

**Algorithm 2:** Pseudo code for stage-II

---

**Inputs :**

$\mathcal{X}$        Source domain (panorama)

$\mathcal{Y}$        Target domain (pinhole image)

$\mathcal{E}_{\text{c}}, \mathcal{E}_{\text{s}}$    Content and style encoders with a deformable Convolution layer

$\mathcal{T}$        Transformer encoder

$\mathcal{G}$        Generator

$\mathcal{D}$        Discriminator

$f_T$       Panorama-to-pinhole image conversion function

$\xi$        Initial parameters for $\mathcal{E}_{\text{c}}, \mathcal{E}_{\text{s}}, \mathcal{T}, \mathcal{G}, \mathcal{D}$, and $f_T$

$\Theta_{\text{ERP}}$    Equirectangular plane offset used in the deformable conv layer

$\Theta_{\varnothing}$    Zero offset

$\theta$        Rotation angle in panoramic augmentation

$S_l$       Number of patches in each $l$-th layer

$N$       Total number of optimization steps

$\eta_n$       Learning rate for $n$-th optimization step

$\alpha$        Panoramic ensemble ratio

$\gamma(\cdot)$     Sinusoidal mapping s.t. $\gamma(a) = \{(\sin(2^{k-1}\pi a), \cos(2^{k-1}\pi a)) | k = 1, ..., K\}$

$\ell(\cdot)$      InfoNCE loss with a positive pair $(\mathbf{v}, \mathbf{v}^+)$, negative pairs $(\mathbf{v}, \mathbf{v}^-)$, and a temperature $\tau$ s.t.

$$\ell(\hat{\mathbf{v}}, \mathbf{v}^+, \mathbf{v}^-) = -\log\left[\frac{\exp(\mathbf{v} \cdot \mathbf{v}^+/\tau)}{\exp(\mathbf{v} \cdot \mathbf{v}^+/\tau) + \sum \exp(\mathbf{v} \cdot \mathbf{v}^-/\tau)}\right]$$

**for** $n = 1$ **to** $N$ **do**

  $\mathcal{B} \leftarrow \{(\text{x}, \text{y}) | \text{x} \sim \mathcal{X}, \text{y} \sim \mathcal{Y}\}$

  **for** $(\text{x}, \text{y}) \in \mathcal{B}$ **do**

    $\text{x}' \leftarrow \text{PanoramicAugment}(\text{x}, \theta)$

    $\mathbf{c}^{\text{x}}, \mathbf{c}^{\text{x}'} \leftarrow \mathcal{E}_{\text{c}}(\text{x}, \Theta_{\text{ERP}}), \mathcal{E}_{\text{c}}(\text{x}', \Theta_{\text{ERP}})$

    $\mathbf{s} \sim \mathcal{N}(0, \mathbf{I})$

    $\hat{\mathbf{y}}^{(\mathbf{0})}, \hat{\mathbf{y}}^{(\mathbf{1})} \leftarrow \mathcal{G}(\mathcal{T}(\mathbf{c}^{\text{x}}, \mathbf{s})), \mathcal{G}(\mathcal{T}(\mathbf{c}^{\text{x}'}, \mathbf{s}))$

    $\hat{\mathbf{y}} \leftarrow \alpha \cdot \hat{\mathbf{y}}^{(\mathbf{0})} + (1 - \alpha) \cdot \text{PanoramicAugment}(\hat{\mathbf{y}}^{(\mathbf{1})}, -\theta)$

    $\mathbf{c}^{\text{x}}, \mathbf{c}^{\text{y}}, \mathbf{c}^{\hat{y}} \leftarrow \mathcal{E}_{\text{c}}(\text{x}, \Theta_{\text{ERP}}), \mathcal{E}_{\text{c}}(\text{y}, \Theta_{\varnothing}), \mathcal{E}_{\text{c}}(\hat{y}, \Theta_{\text{ERP}})$

    $\hat{\mathbf{v}}, \mathbf{v}^+, \mathbf{v}^- \leftarrow \text{PatchSample}_{\text{pos}}(\mathbf{c}^{\hat{y}}), \text{PatchSample}_{\text{pos}}(\mathbf{c}^{\text{x}}), \text{PatchSample}_{\text{neg}}(\mathbf{c}^{\text{x}})$

    $\mathbf{s}^{\text{y}}, \mathbf{s}^{\hat{y}} \leftarrow \mathcal{E}_{\text{s}}(\hat{y}, \Theta_{\varnothing}), \mathcal{E}_{\text{s}}(\hat{y}, \Theta_{\text{ERP}})$

    $\mathcal{L}_{\text{GAN}} \leftarrow \log(1 - \mathcal{D}(\hat{\mathbf{y}})) + \log \mathcal{D}(\mathbf{y})$

    $\mathcal{L}_{\text{df-GAN}} \leftarrow \log(1 - \mathcal{D}(f_T(\hat{\mathbf{y}}))) + \log \mathcal{D}(\mathbf{y})$

    $\mathcal{L}_{\text{NCE,x}}^{\text{cont}} \leftarrow \sum_l \sum_s \ell(\hat{\mathbf{v}}, \mathbf{v}^+, \mathbf{v}^-)$

    $\mathcal{L}_{\text{rand-recon}}^{\text{style}} \leftarrow \|\mathbf{s}^{\hat{y}} - \mathbf{s}\|_1$

    $\mathcal{L}_{\text{recon}}^{\text{img}} \leftarrow \|\mathcal{G}(\mathcal{T}(\mathbf{c}^{\text{y}}, \mathbf{s}^{\text{y}})) - \mathbf{y}\|_1$

  **end**

  $\delta\xi \leftarrow \dfrac{1}{N} \displaystyle\sum_{(\text{x},\text{y}) \in \mathcal{B}} \lambda_{\text{df-GAN}}\mathcal{L}_{\text{df-GAN}} + (1 - \lambda_{\text{df-GAN}})\mathcal{L}_{\text{GAN}}$

        $+ \lambda_{\text{cont}}\mathcal{L}_{\text{NCE}}^{\text{cont}} + \lambda_{\text{style}}\mathcal{L}_{\text{rand-recon}}^{\text{style}} + \lambda_{\text{recon}}\mathcal{L}_{\text{recon}}^{\text{img}}$

  $\xi \leftarrow \text{optimizer}(\xi, \delta\xi, \eta_n)$

**end**

---

## D.3 DIAGRAMS

We additionally illustrate our generation procedure during the training process, shown in Fig. 9. We decompose our training strategy into each training stage, and decompose stage-II into image translation procedure and pinhole image reconstruction. In stage-II, (b) and (c) are trained in parallel. Note that the content encoder, transformer encoder, and generator are shared for (b) and (c) but with different offset values in order to deal with different input types. Also, at the beginning of stage-II, the style encoder and AdaIN layers are re-initialized, in order to learn only the style of target pinhole images.

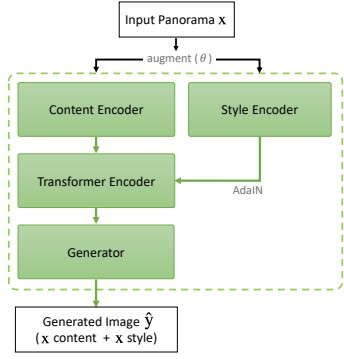

(a) Stage-I, Panorama Reconstruction

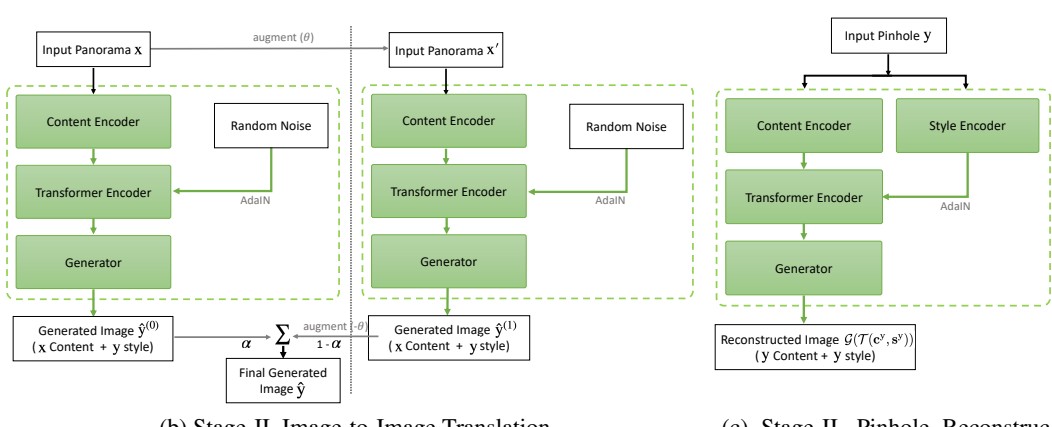

(b) Stage-II, Image-to-Image Translation

(c) Stage-II, Pinhole Reconstruction

Figure 9: **Simplified diagrams** illustrating generation procedure for each stage.

# E ADDITIONAL QUALITATIVE RESULTS

## E.1 INIT DATASET

We include additional qualitative comparisons to other I2I methods, CUT (Park et al., 2020a), FSeSim (Zheng et al., 2021) MGUIT (Jeong et al., 2021), and Instaformer (Kim et al., 2022) on StreetLearn to various conditions of INIT: day→night in Fig. 10, day→rainy in Fig. 11.

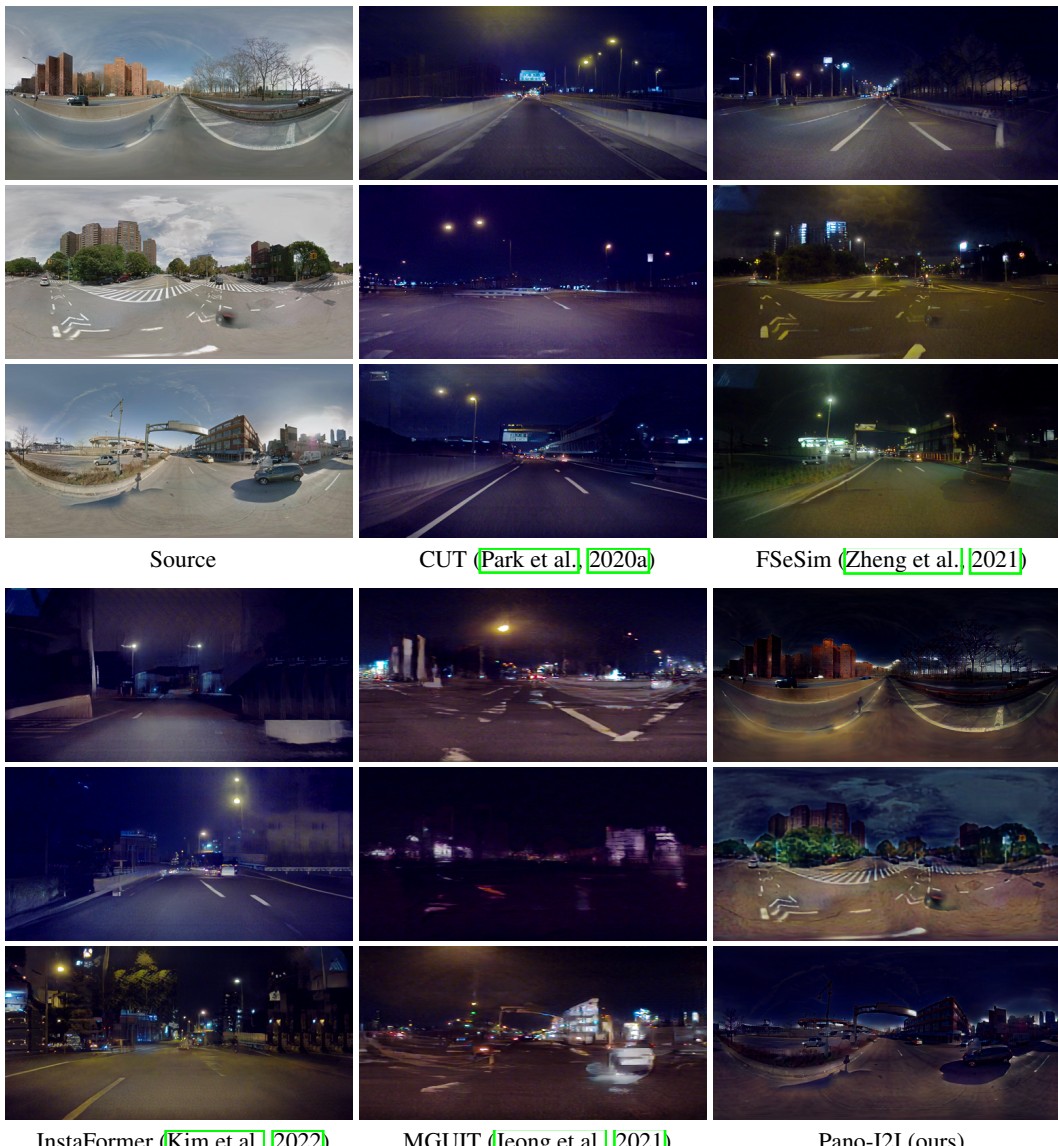

|  |  |  |
|---|---|---|
| Source | CUT (Park et al., 2020a) | FSeSim (Zheng et al., 2021) |

| InstaFormer (Kim et al., 2022) | MGUIT (Jeong et al., 2021) | Pano-I2I (ours) |

Figure 10: **Qualitative comparison** on StreetLearn dataset (Mirowski et al., 2018) (day) to INIT dataset (Shen et al., 2019) (night).

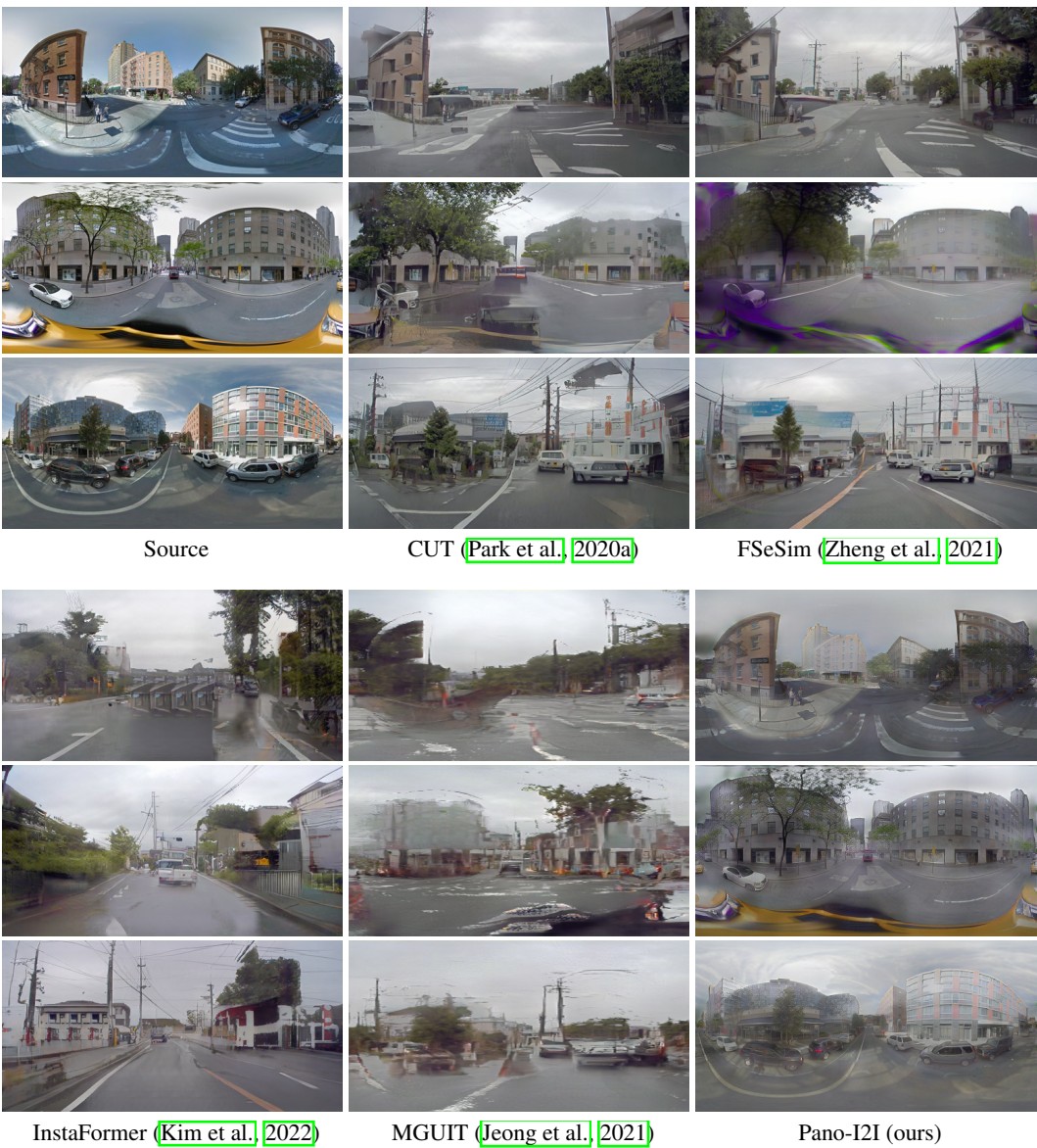

Figure 11: **Qualitative comparison** on StreetLearn dataset (Mirowski et al., 2018) (day) to INIT dataset (Shen et al., 2019) (rainy).

### E.2 Dark Zurich dataset

We visualize additional results of our method on another benchmark for the target domain, including day→twilight in Fig. 12 and day→night in Fig. 13 on StreetLearn and Dark Zurich (Sakaridis et al., 2019). We also provide visual comparisons with other methods (Zheng et al., 2021; Jeong et al., 2021) in Fig. 14.

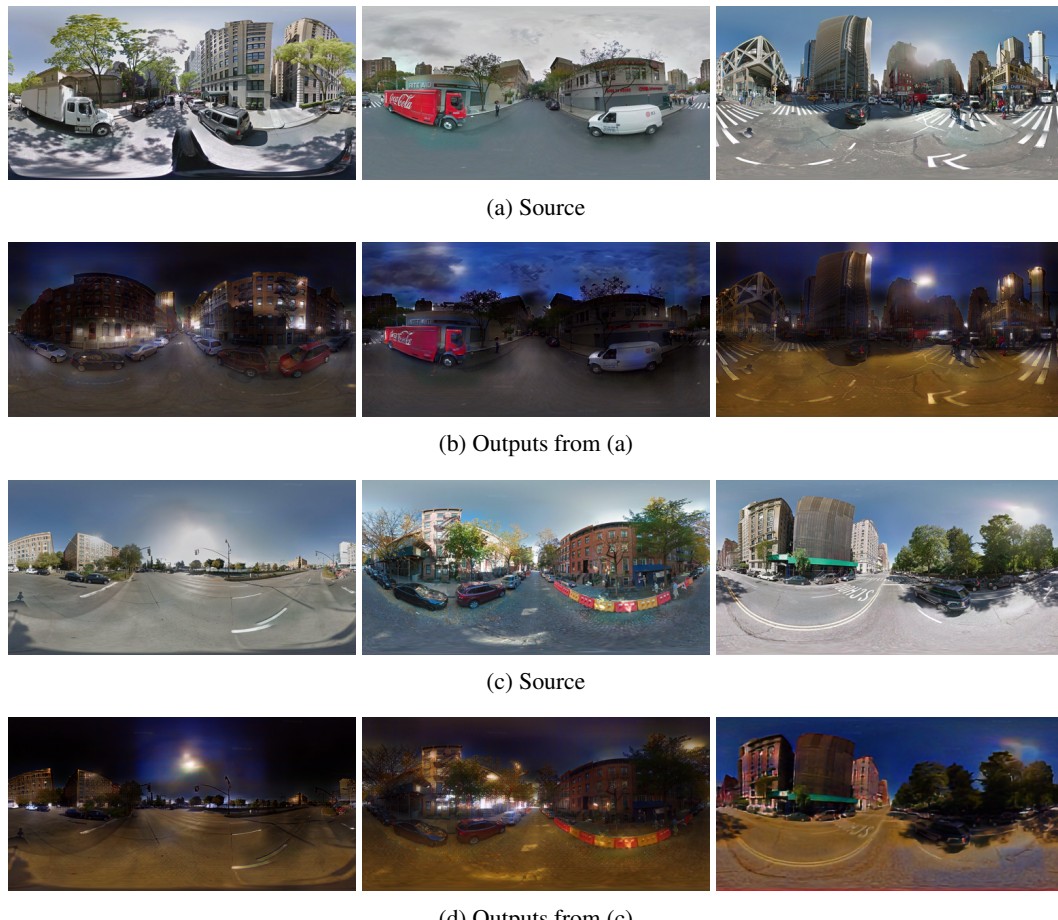

(a) Source

(b) Outputs from (a)

(c) Source

(d) Outputs from (c)

Figure 12: **Qualitative results** from Pano-I2I on StreetLearn dataset (Mirowski et al., 2018) (day) to Dark Zurich dataset (Shen et al., 2019) (twilight).

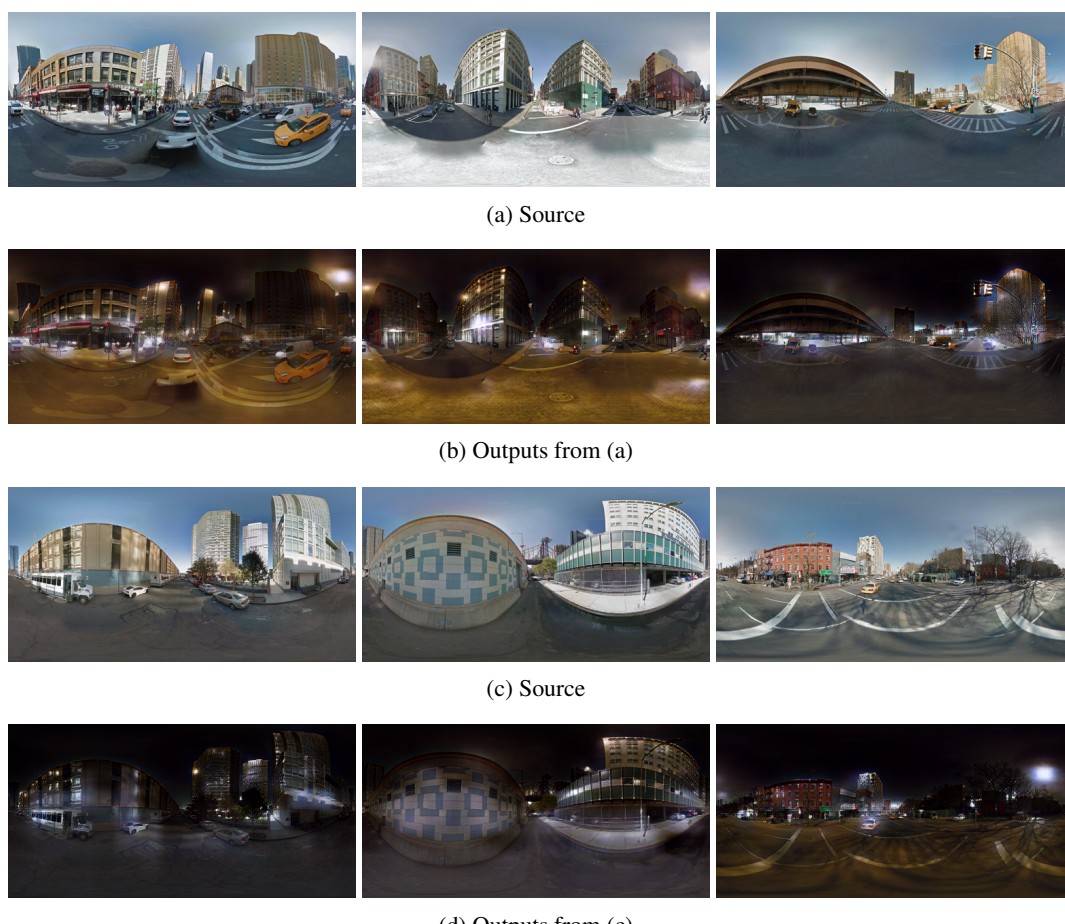

(a) Source

(b) Outputs from (a)

(c) Source

(d) Outputs from (c)

Figure 13: **Qualitative results** from Pano-I2I on StreetLearn dataset (Mirowski et al., 2018) (day) to Dark Zurich dataset (Shen et al., 2019) (night).

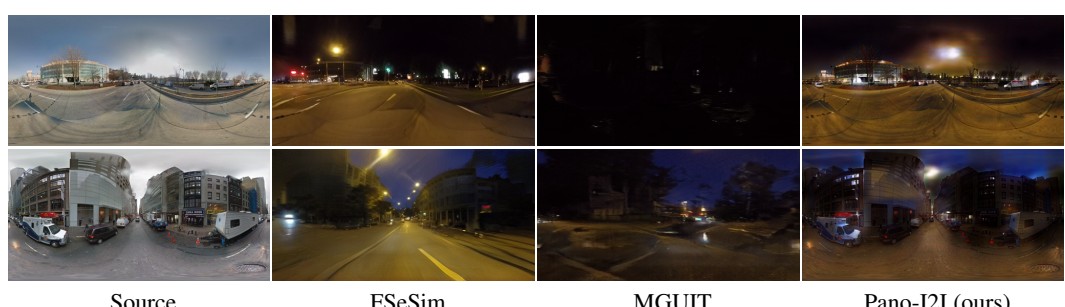

Source        FSeSim        MGUIT        Pano-I2I (ours)

Figure 14: **Qualitative comparison** on StreetLearn dataset (Mirowski et al., 2018) (day) to DZ dataset (Sakaridis et al., 2019) (twilight, night): (top to bottom) day→night, and day→twilight results.

### E.3 MORE COMPARISONS

**SoTA I2I methods and Diffusion-based editing methods.** Since we focus on translating street-view panoramas, we mainly compared with SoTA instance-aware I2I (InstaFormer (Kim et al., 2022), MGUIT (Jeong et al., 2021)). Unlike our method, they additionally use the location of objects, which is an advantage over ours. As mentioned in Section 4.1, we generate and adopt pseudo bbox annotations for training InstaFormer and MGUIT, which provides more advantages to InstaFormer and MGUIT than ours. We additionally compare with other recent I2I methods with public codes: GP-UNIT (Yang et al., 2022), and Decent (Xie et al., 2022) in Fig. 15 and Tab. 7 on StreetLearn to night condition of INIT (day→night). In addition to comparisons with existing GAN-based image-to-image translation works, we also provide qualitative comparisons with recent diffusion-based image editing methods; image, text conditioned I2I method: IP-Adapter (Ye et al., 2023), text conditioned I2I method: T2IAdapter (Mou et al., 2023).

Both GP-UNIT and Decent fail to preserve panoramic structures and show collapsed objects. We observed that the results of IP-Adapter fails to capture structure from the source image carefully. T2IAdapter shows relatively successfully preserves the overall structure of the input panorama image, yet the results still show limited performance; the shape of the road seems distorted and the model generates undesired objects.

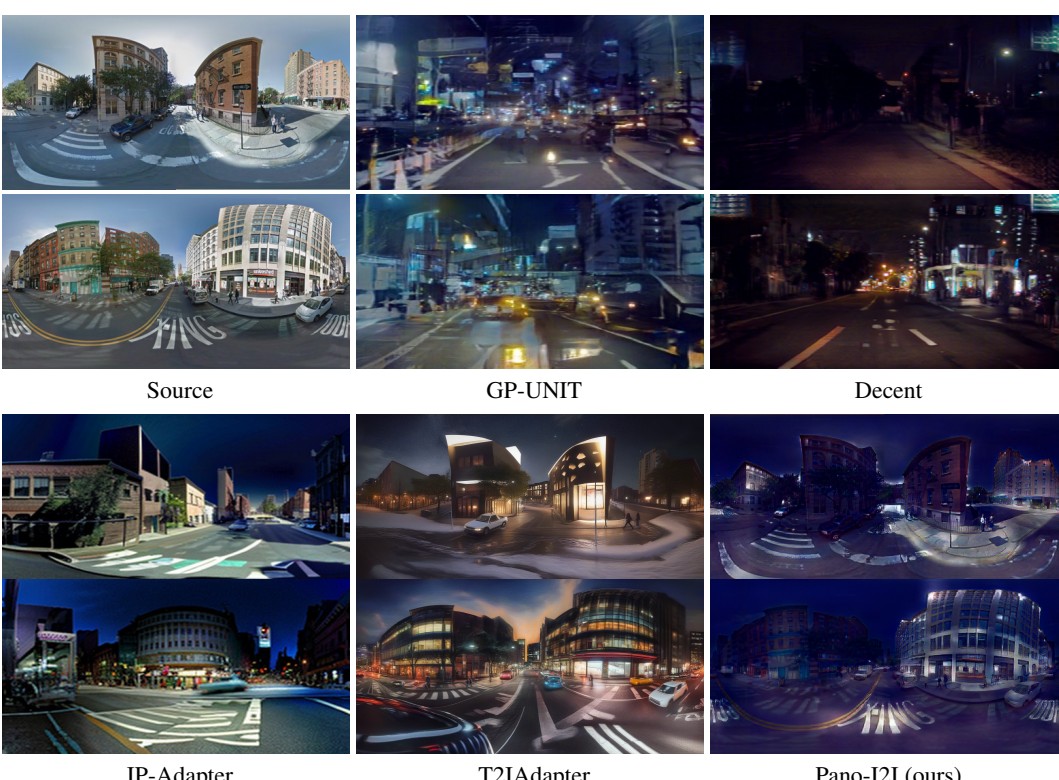

Figure 15: **More qualitative comparisons on daytime→night.**

Table 7: **More quantitative evaluation on daytime→night.**

| Metric | Day→Night | |
| --- | --- | --- |
| | FID↓ | SSIM↑ |
| GP-UNIT (Yang et al., 2022) | 225.6 | 0.153 |
| Decent (Xie et al., 2022) | 180.8 | 0.177 |
| IP-Adapter (Ye et al., 2023) | 165.3 | 0.141 |
| T2IAdapter (Mou et al., 2023) | 183.4 | 0.182 |
| Pano-I2I (ours) | **94.3** | **0.417** |

**Pretrained I2I model based on semantic masks.** We conduct an experiment with PITI (Wang et al., 2022), which aims to generate diverse images based on a semantic map with pretrained models. Since StreetLearn dataset does not contain ground-truth semantic masks, we apply Segment Anything (Kirillov et al., 2023) to obtain semantic masks. As shown in Fig. 16, although PITI maintains the panoramic structure, it cannot capture the semantic information from the source, and outputs unrelated content. Also, it generates unseen objects and unrelated content in the background, as seen in the both results. In contrast, our results successfully capture the structure and semantic information from the source, while translating into desired conditions (day→rainy and day→night, respectively). Note that PITI only takes a source as the input, and it is unable to control the style.

| Source | Semantic mask | PITI (Wang et al., 2022) | Pano-I2I (ours) |
|---|---|---|---|

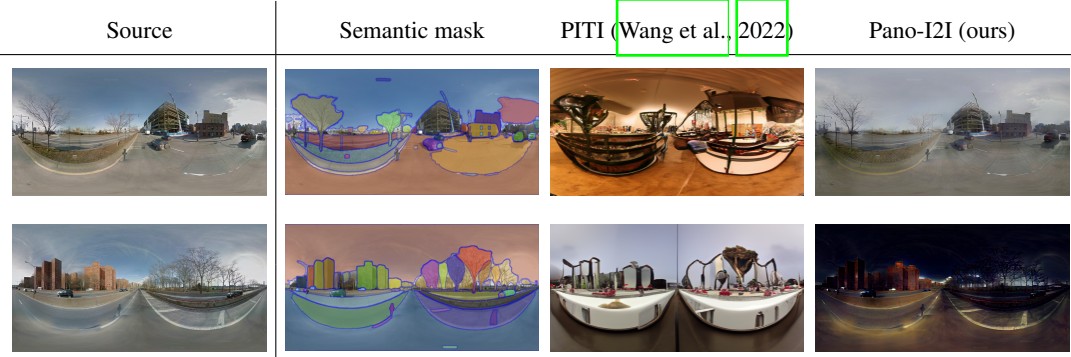

Figure 16: **Qualitative comparisons with PITI (Wang et al., 2022).** Note that we obtain semantic masks using Segment Anything (Kirillov et al., 2023).

**Style transfer methods.** We also conduct experiments with SoTA style transfer methods, EFDM (Zhang et al., 2022c) and DTP (Kim et al., 2021). For EFDM, we conduct inference using panoramic source and pinhole target using pretrained models, and for DTP, we optimize the model using panoramic source and pinhole target. The source and target and the results are shown in Fig. 17. In the results, EFDM fails to stylize in the first row, since it cannot capture the semantic relation between source and target due to the large distribution gap. In the second row, the generated result of EFDM fails to maintain the structure from the source (buildings, trees), and simply changes the global style to be very dark. DTP shows much better results in terms of style and content preservation, however, the results exhibit the emergence of unrealistic patterns or objects that were not present in the source, and fail to preserve structure from the source (trees, lanes) in the second row.

| Source | Target | EFDM (Zhang et al., 2022c) | DTP (Kim et al., 2021) | Pano-I2I (ours) |
|---|---|---|---|---|

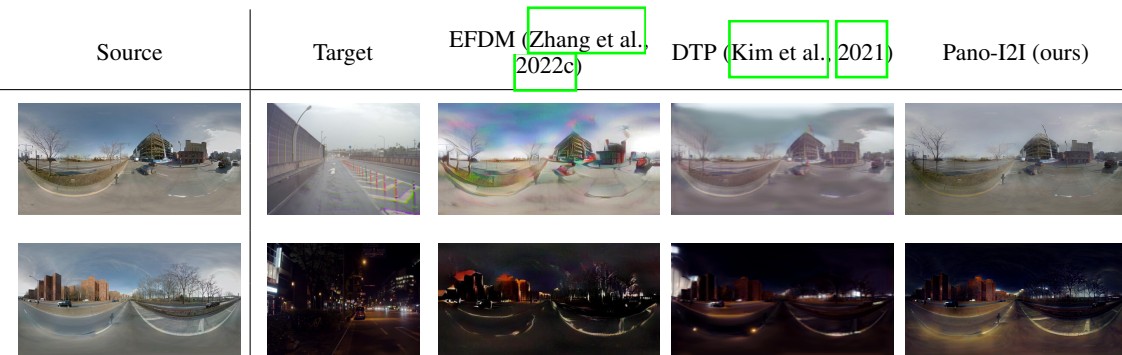

Figure 17: **Qualitative comparisons with style transfer methods, EFDM (Zhang et al., 2022c) and DTP (Kim et al., 2021), on daytime→rainy and daytime→night.**

**Cubemap projection with existing SoTA I2I method.** Since image translation for panorama inputs has not been explored yet, in the main paper, we have compared with prior SoTA I2I methods after training them with panoramic source and pinhole target as in our setting. In order to further substantiate the necessity of our framework and facilitate a more convincing demonstration, we conduct an experiment: combining cubemap projection-commonly used in panorama image modeling-with existing SoTA I2I method, FSeSim (Zheng et al., 2021). In this experiment, we replace the type of input panoramas from ERP into cubemaps, which has six faces and is known for less structure distortion than ERP. We first project an panorama input into six cubmaps, process them into the FSeSim network, and re-project them into ERP to obtain the final result. We train the network using StreetLearn dataset as the source and INIT (rainy) dataset as the target.

Fig. 18 illustrates the results; we denote FSeSim with cubemap projection as FSeSim-cube. While the results of (b) FSeSim-cube capture some panoramic structure, it displays inconsistent structure, artifacts in the images, and some objects (buildings) are disappeared or changed into another objects (trees). This is because the panorama is decomposed into six faces; some of the cubemaps have unrelated content from the pinhole target (*e.g.,* sky only, road only), thus containing large distribution gap each cubemap. Therefore, each cube fails to maintain the original content from the source, and follows the structurural distribution from the pinhole target, which is similar to the results of vanilla FSeSim (Fig. 4 in the main paper). Compared to this, our method (c) succeeds in recovering detailed structure and content of the source without discontinuity, as we desired. As discussed in Section 3 and Section 4, we argue that large view differences and distribution gap make it difficult to disentangle the content and style for the source and the target while maintaining the structure from the source.

|  (a) Source | (b) FSesim-cube | (c) Pano-I2I (ours) |
| --- | --- | --- |

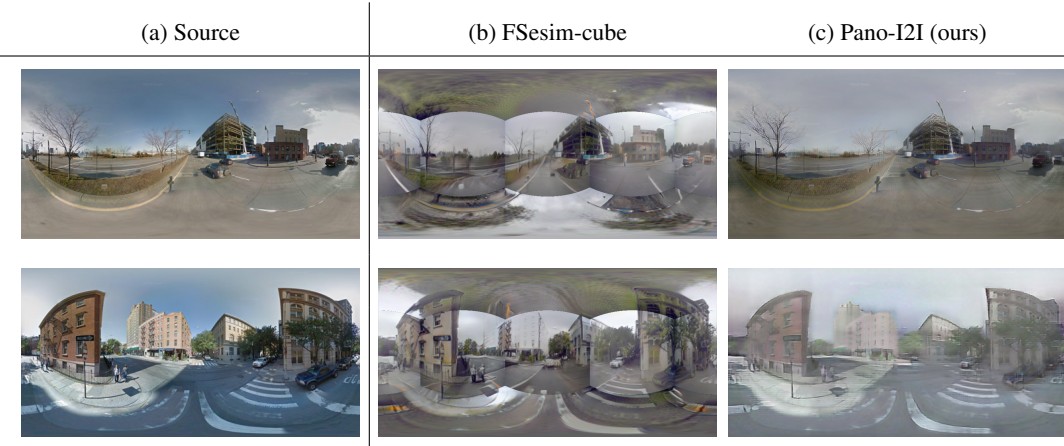

Figure 18: **Qualitative comparisons with FSeSim with cubemap projection** on StreetLearn to INIT (rainy).

### E.4 VISUALIZATION ON OBJECT DETECTION RESULTS

We visualize results of object detection in Fig. 19 on translated panoramas on StreetLearn to night condition of INIT (day→night), using YOLOv5. As seen, our model excels in preserving objects from the source, compared to others.

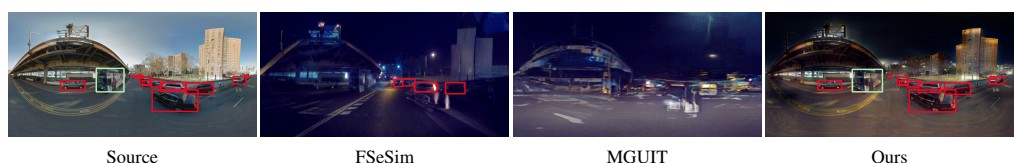

| Source | FSeSim | MGUIT | Ours |
| --- | --- | --- | --- |

Figure 19: **Visualization on object detection results; red box: car, green box: bus.**

## F    DETAILS OF USER STUDY

We conduct a user study to compare the subjective quality, shown in the main paper. We randomly select 10 images for each task (sunny→night, sunny→rainy) for INIT dataset, compared with CUT (Park et al., 2020a), MGUIT (Jeong et al., 2021), FSeSim (Zheng et al., 2021), and InstaFormer (Kim et al., 2022). We request 60 participants to evaluate the quality of synthesized images, content relevance, and style relevance considering the context. In particular, each instruction is as follows:

(1) Image quality. Given a row of 5-images, please select the index of image that has the best image quality.

(2) Content relevance. Given a row of 5-images and a content image, please select the index of image that has the most similar structure and content with A single content image.

(3) Style relevance considering the context. Given a row of 5-images and a content image and a style image, please select the index of image that has the most similar global style (weather, time, color..) with the style image. Note that the images should have same content from the content image.

# G LIMITATIONS

Although our method shows outstanding performance on various benchmarks, our method inherits a problem in style relevance. Specifically, as observed in Fig. 20, Pano-I2I is robust for preserving the panoramic structure, while sometimes struggling to represent the desired style for some samples. There is still room for performance improvement, so fostering research is needed.

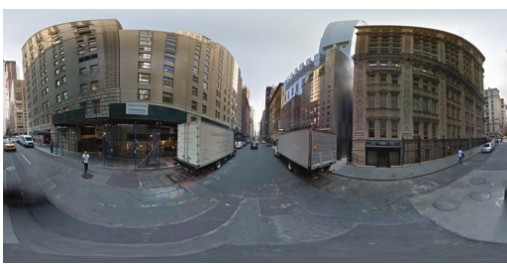 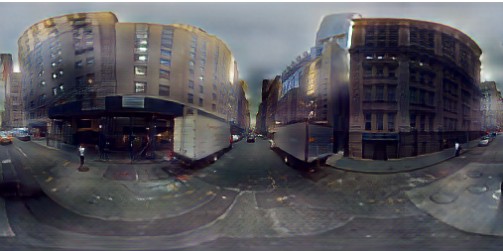

Input                                    Output (night)

Figure 20: **Failure case** of Pano-I2I on StreetLearn dataset (Mirowski et al., 2018) (day) to INIT dataset (Shen et al., 2019) (night).