# OpenReview forum: "Unpaired Panoramic Image-to-Image Translation Leveraging Pinhole Images"
_ICLR.cc/2024/Conference — Submitted to ICLR 2024_

### Official Review · Reviewer_7NBx · 2023-10-21

**Soundness:** 2 fair
**Presentation:** 3 good
**Contribution:** 2 fair
**Rating:** 5
**Confidence:** 4

**Summary:**

This paper proposes a new approach for unpaired panoramic image-to-image translation, which addresses the challenges of modifying naive 360-degree panoramic images using readily obtainable pinhole images as the target domain for training. The authors introduce a new model that leverages spherical position embedding and sphere-based rotation augmentation to address discontinuities at the panorama edges. The proposed method is evaluated on several datasets and compared with state-of-the-art methods, demonstrating superior performance in terms of both quantitative metrics and visual quality.

**Strengths:**

The proposed method is well-motivated and addresses the problem in the field of panoramic image processing. The authors provide a clear and detailed description of the model architecture and training procedure, which makes it easy to reproduce the results. The experimental evaluation is thorough and includes both quantitative and qualitative analysis, which demonstrates the effectiveness of the proposed method. The use of spherical position embedding and sphere-based rotation augmentation is a novel contribution that significantly improves the performance of the model.

**Weaknesses:**

1: Enhancing the motivation behind the task is crucial; specifically, showing practical applications of Panoramic Image-to-Image Translation would be great.

2: How can the proposed method  help downstream applications such as AR/VR and driving? It would be great to see more results and discussions on this part.

3: Delving into the computational aspects, including time, memory, and overall efficiency, and comparing these metrics with other techniques, would add substantial value to the discussion.

4: It appears that the Ensemble technique and SPE deform conv have a relatively minor impact on the overall pipeline. Given this, I recommend conducting an ablation study, progressing step by step, as demonstrated in Table 1 of the StyleGAN paper ("A Style-Based Generator Architecture for Generative Adversarial Networks") to better presents the contribution of each module.

**Questions:**

Please see weaknesses.

---

> ### Author Response · Authors · 2023-11-20
> **Official Comment to Reviewer 7NBx**
>
> We highly appreciate your constructive comments. Our response is shown below. If our responses do not adequately address your concerns, please let us know and we will get back to you as soon as possible.
>
> ### Practical applications
>
> Since existing I2I methods on outdoor scenes are often adopted for data augmentation for autonomous driving, our task can be directly applied for data augmentation for adverse conditions in AR/VR, autonomous driving, and street view services.
> To demonstrate this, we have visualized the results of object detection in Appendix G on translated panoramas using YOLOv5, showing our method excels in preserving objects.
> Also, some of the motivations from our work can be also adopted in another tasks, *e.g.,* domain adaptation for semantic segmentation.
>
> ### More analysis on ablations
>
> We apologize that we could not conduct additional ablation studies for our SPE and ensemble technique at this time, due to the lack of training resources and time. We will add this in the final version.
>
> We believe that SPE helps preserve structural continuity, since SPE is applied at feature-level. The ensemble technique is conducted at image-level, so it works for style continuity at the left-right boundaries of an image.
>
> ### Comparisons on model complexity
>
> We compare an inference time and the number of the parameters below. It shows a comparison on inference time per 100 images and the number of parameters. While ours has more parameters since our method is based on ViT, it achieves competitive time consumption.
>
> |  | CUT | FSeSim | MGUIT | InstaFormer | Ours |
> | --- | --- | --- | --- | --- | --- |
> | Time per 100 images (sec.) | 40.36 | 40.19 | 39.20 | 43.02 | 39.58 |
> | # of param (M.) | 14.7 | 14.3 | 49.3 | 75.5 | 74.3 |

---

> > ### Comment · Reviewer_7NBx · 2023-11-21
> > **Thank you for your response**
> >
> > Thank you for providing insights into the practical applications of visual recognition tasks and presenting results.Also its good to see the inclusion of comparisons regarding complexity.
> >
> > The response has mostly addressed my concerns, and I am inclined to maintain my rating as 5.

---

### Official Review · Reviewer_4pKv · 2023-10-23

**Soundness:** 3 good
**Presentation:** 2 fair
**Contribution:** 2 fair
**Rating:** 5
**Confidence:** 2

**Summary:**

This work designed a new method for panorama style transfer. Unlike natural images, panorama has several differences, such as distortion. To bridge the gap between image-to-image translation approaches and panoramic images, this submission proposed distortion-aware panoramic modeling techniques and distortion-free discriminators for this problem. Experimental results on some benchmarks show the effectiveness of the proposed method.

**Strengths:**

- Though image-to-image translation (style) is a typical problem, this paper focuses on a new setting, style transfer for panorama. I believe there would be a lot of applications of panoramic image style transfer in XR.
- Given the differences between panorama and natural photos, this paper designed some modules for panorama, and experiments show improvement over previous methods.

**Weaknesses:**

- The writing can be further improved and polished to make the submission stronger. Some equations such as EQ(8)(9) confuse me, as there are so many symbols and notations. It would be better to include more insights and explanations in the context rather than listing equations. Also, Fig. 3 is hard to follow and I would suggest the authors re-organizing it. Which parts are Stage I? Which parts are Stage II? What do notaitons mean in Fig.3? It would be better to make the figure self-contained.
- Baselines. The tasks of Day-to-Night and Day-to-Rain are very similar to unpaired image-to-image translation tasks in CycleGAN and many follow-up works. This task is also related to image style transfer, as there is a content panorama and a style pinhole image. Could you compare your method with some style transfer works? Also, the baseline results in Fig.4 are too bad, and the shapes are even not preserved. I would suggest double-checking the baseline results and comparing to some up-to-date image-to-image translation works with diffusion models.

**Questions:**

- Could you show some failure cases of the proposed method?
- The submission only shows results on two types of translation, i.e., Day-to-Night, Day-to-Rain. This makes it difficult to determine whether the proposed method is general. Could you show more results on more styles?
- Compared to pretrained methods. One motivation of this paper is it is difficult to collect paired data for training. Could you discuss the pros and cons compared with methods with pretraining [1]?
[1] Pretraining is All You Need for Image-to-Image Translation. 2022

---

> ### Author Response · Authors · 2023-11-20
> **Official Comment to Reviewer 4pKv**
>
> We highly appreciate your constructive comments. Our response is shown below. If our responses do not adequately address your concerns, please let us know and we will get back to you as soon as possible.
>
> ### Writing, equations, notations
>
> We agree that equations and notations are complicated. We will further elaborate our writing.
> We kindly refer the reviewer to CUT[ECCV'20], FSeSim[CVPR'21] for Eq(8) and Eq(9), as we borrowed the equations from those literature. Also, we kindly remind that we have provided the meaning of notations in Appendix A.
> Also, please refer to the pseudo-code illustrating the training algorithm of each stage in Appendix D.1 and D.2 for more details.
>
> ### Fig.3 is complex
>
> We noticed that the architecture (Fig. 3) that illustrates Stages 1 and 2 could lead to a misunderstanding. To further reflect your comment, we added simplified diagrams for each stage in Appendix D.3.
>
> ### More comparisons with other methods (style transfer, Diffusion models)
>
> Thank you for your constructive comment. Following this, we conducted experiments and added comparisons with SoTA style transfer methods (EFDM, DTP) in Appendix E.3.
> Our method clearly outperforms other methods in terms of style translation regarding the context and content preservation, since they fail to capture semantic relation between source and target due to their large domain gap.
>
> Also, we kindly remind you to Appendix E.3 which includes qualitative and quantitative comparisons with more recent I2I methods based on GANs (GP-UNIT, Decent) and recent image editing methods based on Diffusion models (IP-Adapter, T2IAdapter).
>
> ### Suggestion for double-checking the baseline results
>
> Thanks for the suggestion. We conduct experiments using default hyper-parameter settings for other methods. Also, as our architecture design is based on one of the comparison methods (InstaFormer), we are convinced that the results of comparison methods are not wrong.
>
> ### Failure cases
>
> Please refer to Appendix G to find the limitations and failure cases.
>
> ### More translation results except for day-night and day-rainy
>
> We kindly remind you that our results on the Dark Zurich dataset (night, twilight) are in the main paper and Appendix.
>
> ### Comparison with pretrained methods
>
> We respectfully remind you to see Appendix E.3; we have also provided comparisons with recent diffusion-based image editing methods that are based on pretrained networks.
> Following your comment, we also added qualitative comparisons with the suggested work (PITI) in Appendix E.3. For generating semantic masks, we adopt Segment Anything.
>
> As shown in Fig.15 in Appendix, although PITI has strength in maintaining the panoramic structure compared to existing I2I methods, it cannot capture any semantic information from the source, and outputs unrelated content. Also, it generates unseen objects and unrelated content in the background, as seen in both results.
> In contrast, our results successfully capture the structure and semantic information from the source, while translating into desired conditions (day→rainy and day→night, respectively).
> Note that PITI only takes a source as the input, and it is unable to control the style.

---

> ### Author Response · Authors · 2023-11-22
> **Hope to get the feedback.**
>
> Dear reviewer 4pKv,
>
> Sorry for bothering you. We genuinely appreciate the time and effort you've invested in our paper. As our rebuttal and the revision of our paper have been submitted for a while, we do want to know whether our explanations have settled the concerns. We are eager to have further discussions, so please let us know if you have additional feedback.
>
> Best,
> Authors of submission 581.

---

### Official Review · Reviewer_HC91 · 2023-10-28

**Soundness:** 3 good
**Presentation:** 3 good
**Contribution:** 2 fair
**Rating:** 3
**Confidence:** 4

**Summary:**

In this work, the authors propose a new task, to learn the mapping from unpaired panoramic source to non-panoramic target. They propose a new I2I model for it. The experimental results show the effectiveness of their model.

**Strengths:**

This paper is well-written and easy to follow. The new task is interesting. They have done comparison and ablation studies to validate their proposed components.

**Weaknesses:**

However, there the following concerns to prevent me vote for acceptance.

The first one is the unfair comparison, and some examples might not be sufficient supports. In Fig. 2, the FSeSim is designed for planar images, not panoramic images. If directly using FSeSim in panoramic images, it’s unfair comparison. The authors should compare the style transfer ability by using cube map project, which transfers the sphere data into six faces. And FSeSim is used on each face image and finally combined. In this way, the comparison is fair. Similar in Fig. 4, compared SOTA are designed for planar images, and it’s unfair in this directly inference.

The second one is the used projection, where the better one is cube map, not ERP. Using cube map projection, the main things is to make consistency between six face, even for the existing method.

**Questions:**

See weakness.

---

> ### Author Response · Authors · 2023-11-20
> **Official Comment to Reviewer HC91**
>
> We highly appreciate your constructive comments. Our response is shown below. If our responses do not adequately address your concerns, please let us know and we will get back to you as soon as possible.
>
> ### Unfair comparisons
>
> Regarding that, this is the first work on both 1) image translation for panoramas, and 2) image translation for panoramas leveraging pinhole images as target, we believe that our comparison pipeline (training existing SoTA I2I methods with panoramic source and pinhole target) was the best choice.
> Also, there is no prior work on image translation using cubemap projection.
>
> ### Cubemap projection
>
> Thank you for your suggestion. Following it, we conducted an experiment using cubemap projection for FSeSim, shown in Appendix E.3.
> While it shows better results than original FSeSim, unfortunately, we observed that this setting did not converge as we desired. Since each cubemap contains a different content distribution and it also shows a large discrepancy compared to the pinhole image source, some cubemaps naively become to follow the target distribution without considering the context of the source. In addition, it contains artifacts and undesired patterns in the output image, and shows structural discontinuity and style inconsistency at the boundaries.
>
> Additionally, after projecting the generated cubemaps into an equirectangular, the structural discontinuity and style difference at the boundaries seem worse than the original FSeSim.
>
> In addition to the observation from the experimental results, we respectfully disagree that cubemap projection is optimal for image translation for two reasons.
> Cubemap projection could be an optimal design choice if both source and target are panoramic images since they have the same content distribution. However, in our task, *i.e.,* *generating panoramic outputs with panoramic source and pinhole target images*, the source and target have critically different content distribution.
> After cubemap projection for a full-FoV panorama image, some of the projected cubemaps exhibit a content distribution that differs from that of the target pinhole image (i.e., containing only sky or road), thus forcing output cubmaps to follow the distribution from pinhole target images (we visualize this case in Appendix). If we train existing I2I methods with those cubemaps as source and pinhole images as a target, where the source and target often contain unrelated content distributions, it would fail to align with the desired objectives and tend to cause to mode collapse.
>
> Also, it brings out another challenge of maintaining both structural continuity and global style consistency with the desired style, which has not been explored yet.

---

> ### Author Response · Authors · 2023-11-22
> **Hope to get the feedback**
>
> Dear reviewer HC91,
>
> Sorry for bothering you. We genuinely appreciate the time and effort you've invested in our paper. As our rebuttal and the revision of our paper have been submitted for a while, we do want to know whether our explanations have settled the concerns. We are eager to have further discussions, so please let us know if you have additional feedback.
>
> Best,
> Authors of submission 581.

---

### Official Review · Reviewer_jBQQ · 2023-10-31

**Soundness:** 3 good
**Presentation:** 3 good
**Contribution:** 3 good
**Rating:** 5
**Confidence:** 5

**Summary:**

This work proposes a panoramic I2I task by translating panoramas with pinhole images as a target domain. In particular, a versatile encoder and distortion-free discrimination are designed to efficiently bridge the large domain gap between panoramic and pinhole images. To address the discontinuities at the panorama edges, the strategies of spherical position embedding and sphere-based rotation augmentation are proposed. Experimental results demonstrated that the proposed work significantly outperforms the previous methods.

**Strengths:**

+ To my knowledge, this is the first work that leverages pinhole images to guide the panoramic I2I task. It well addresses two challenges: directly applying conventional I2I methods cannot perceive the specific geometric distortion in panoramic images; panoramic image translation is the absence of sufficient panorama datasets covering diverse conditions.
+ The supplementary materials allow quite convincing supports for this work. More implementation details, network architectures, comparison evaluations, and deep analyses of experiments are offered.
+ The experiments are comprehensive, and some comparison samples look promising.

**Weaknesses:**

- While this work provides the first step for the panoramic I2I task by translating panoramas with pinhole images as a target domain, the novelty seems kind of limited from my perspective. For example, the adaptation from source (pinhole) dataset with annotated label to target (panoramic) dataset (Mutual Prototypical Adaptation) has already explored in "Bending Reality: Distortion-aware Transformers for Adapting to Panoramic Semantic Segmentation"; the similar panoramic modeling strategy in encoders has been exploited in "Disentangling Orthogonal Planes for Indoor Panoramic Room Layout Estimation with Cross-Scale Distortion Awareness" and "Spherical Convolution"; the discontinuity elimination strategy is also studied in "Cylin-Painting: Seamless 360° Panoramic Image Outpainting and Beyond with Cylinder-Style Convolutions", "Diverse plausible 360-degree image outpainting for efficient 3dcg background creation", "Spherical Image Generation From a Few Normal-Field-of-View Images by Considering Scene Symmetry", etc. Please briefly review the above literature and discuss your customized contribution beyond these baselines.
- In Figure 3, the authors claimed that Stage I only learns to reconstruct the panorama source, and Stage II learns panorama translation. However, it is hard to discriminate the difference between stage I and stage II in the figure.
- For the architecture, the motivation of using a shared content encoder and a shared style encoder to learn both panoramic images and pinhole images is ambiguous and unclear, given the context in which their domains significantly differ.
- Did the comparison method retrain on the panoramic images? More implementation details are expected to be provided.

**Questions:**

Could the pinhole-image-guided panoramic method be extended to other tasks, such as panoramic image inpainting and outpainting?

---

> ### Author Response · Authors · 2023-11-20
> **Official Comment to Reviewer jBQQ (1/2)**
>
> We highly appreciate your constructive comments. Our response is shown below. If our responses do not adequately address your concerns, please let us know and we will get back to you.
>
> ### The novelty seems kind of limited from my perspective.
>
> We would like to clarify that our method differs from the mentioned works and additional technical contributions.
>
> > Domain adaptation for segmentation ("Bending Reality")
>
> Such works on domain adaptation for segmentation require class-wise labels since the loss functions and training strategies are mostly based on class-wise information, and they focus on aligning the distributions between source and target class-wise.
> Whereas our task does not exploit any class label or bounding boxes within an image, we only use the information about per-sample conditional annotations (e.g., daytime, night, rainy).
>
> Another difference is that one main point for the mentioned task is the **style similarity** for the segmentation map for the source and target, as seen in ``Learning to Adapt Structured Output Space for Semantic Segmentation[CVPR'22]'''s mention that "While images may be very different in appearance, their outputs are structured and share many similarities."
> In contrast, our task poses challenges as the source and the target have different styles and different distortions. Therefore, one of the main points of our work is to disentangle content and style for source and target and generate a translated image considering left-right structural- and style-continuity. To do so, we have proposed distortion-free discrimination and adopted several techniques (spherical PE, rotational augmentation, and fusion).
> For those reasons, the approaches from such works (prototype, clustering, adversarial learning) cannot be directly adopted into image translation tasks.
>
> > Similar panoramic modeling strategy in encoders ("Disentangling Orthogonal Planes for Indoor Panoramic Room Layout Estimation", "Spherical Convolution")
>
> As mentioned in our paper, we indeed adopted the deformable convolution motivated by existing works (PanoFormer, PAVER).
> Though our deformable conv is inspired by PAVER, the motivation is significantly different. 1) We handle panorama source and pinhole target inputs using heterogeneous encoders but with different offsets for deformable convolutions and different PE techniques to reduce the geometrical gap. Still, we process them via a shared network to align them in a shared representation space, which helps to disentangle content and style better.
> In addition, this approach is efficient as the same encoders learn the compact representation that explains both types of images.
> We would like to emphasize that there has been no prior research utilizing a shared network while adjusting offsets based on whether it is the source or target, considering distortion in both panorama and pinhole within the content encoder and discriminator.
>
> > Discontinuity elimination strategy (”Cylin-Painting”, Diverse plausible 360-degree image outpainting for efficient 3dcg background creation,'' ``Spherical Image Generation From a Few Normal-Field-of-View Images by Considering Scene Symmetry'')
>
> The paper ``Diverse plausible 360-degree image outpainting for efficient 3dcg background creation'' employs a Transformer-based approach to address left-right discontinuity through circular padding and circular inference, which is a commendable method. However, in our work, we introduce the absolute Spherical Patch Embedding (SPE) as a straightforward solution. Additionally, we extend our focus beyond panorama images by handling pinhole images with a shared network. To achieve this, we divide the Patch Embedding (PE) into Spherical PE and fixed PE, applying them separately to the source and target. Notably, we share the parameters of the Transformer between the two types of inputs. In contrast, the method proposed in this paper is applicable **only to panorama images** due to variations in patch numbers for a panorama and a pinhole image and even involves more parameters for the Transformer than our SPE.
>
> In the case of "Cylin-Painting," the paper provides an analysis solely on Sinusoidal PE and Padding PE, but not Spherical PE.
>
> Finally, the last paper also employs circular padding.
>
> In addition, we address other technical novelty as below:
>
> > Additional technical novelty
>
> Also, we emphasize our differentiable rectilinear projection is prominent to deal with the distortion and distribution differences between source/target, aligning them for the discriminator and adversarial losses.
>
> In addition, our motivation for the ensemble is to specifically resolve style discrepancy at boundaries in panoramic I2I, considering rotational equivariance in panoramas(Eq.5), which has not been explored yet, compared with existing works that ensemble different types of projections (UniFuse, BiFuse, OmniFusion) or tangent images into a panorama (360MonoDepth), often requiring additional blending techniques.

---

> ### Author Response · Authors · 2023-11-20
> **Official Comment to Reviewer jBQQ (2/2)**
>
> ### Fig.3 is complex
>
> We noticed that the architecture (Fig. 3) that illustrates Stages 1 and 2 could lead to a misunderstanding. To further reflect your comment, we added simplified diagrams for each stage in Appendix D.3.
> Also, please refer to the pseudo-code illustrating the training algorithm of each stage in Appendix D.1 and D.2 for more details.
>
> ### Details on shared encoders
>
> We noticed that Fig.3 appears to be complicated and may have led to your misunderstandings.
>
> We handle panorama source and pinhole target inputs using heterogeneous encoders but with different offsets for deformable convolutions and different PE techniques to reduce the geometrical gap. For pinhole inputs, we set zero for the offset values (Θ∅), so it works as vanilla convolution when patchfying the inputs. In contrast, for panorama inputs, we set ΘERP for the offset.
> Also, it should be noted that we have a style encoder only for the target style in Stage 2 with Θ∅, while Stage 1 only contains the style encoder with ΘERP for source style (daytime).
> More specifically, we can divide Stage 2 into two processes: (1) image translation with a pinhole target and panoramic source, (2) image reconstruction with the pinhole image as source and target.
> In (1), we use randomly generated style code from Gaussian distribution and process it into AdaIN, so in this process, a style encoder is not used.
> In (2), we aim to reconstruct pinhole image input. Thus, we extract the style from the pinhole input using a style encoder, and process it into AdaIN.
>
> Thus, we re-initialize our style encoder at the beginning of Stage 2; we do not share the style encoder across different styles. We clarified this in Appendix D., and we added simplified diagrams describing each training stage in Appendix D.3, so please refer to Figure 9 in Appendix D.3.
>
> ### Details on training comparison methods
>
> As you mentioned, we retrain the comparison methods using panoramas as the source and pinhole images as a target. We conduct experiments using default hyper-parameter settings for other methods. We clarified this in Experiments section.
>
> ### Future work
>
> We think our network cannot be straightforwardly adopted to inpainting or outpainting, but we believe our network can be adopted as a baseline for such image editing tasks. Also, our framework can be directly applied to some downstream applications, including objection detection and segmentation. For example, we have visualized the results of object detection in Appendix E.4 on translated panoramas using YOLOv5, showing our method excels in preserving objects.

---

> > ### Comment · Reviewer_jBQQ · 2023-11-22
> > **Thanks for the Response**
> >
> > Thanks for the detailed response, which addressed some of my concerns. Please further incorporate the above reply into the manuscript. However, the contributions of the panoramic model strategy and the discontinuity elimination strategy still seem weak to me. Therefore, I would like to keep the original rating.

---

> ### Author Response · Authors · 2023-11-22
> **Hope to get the feedback**
>
> Dear reviewer jbQQ,
>
> Sorry for bothering you. We genuinely appreciate the time and effort you've invested in our paper. As our rebuttal and the revision of our paper have been submitted for a while, we do want to know whether our explanations have settled the concerns. We are eager to have further discussions, so please let us know if you have additional feedback.
>
> Best,
> Authors of submission 581.

---

### Author Response · Authors · 2023-11-20
**General Response**

We would first like to express our gratitude for the insightful reviews and helpful suggestions provided by the reviewers. We are greatly encouraged by their assessment of our new problem setting and motivation as interesting (HC01, 4pKv, 7NBx), challenges are well described (jBQQ) with comprehensive experiments (jBQQ) and significant developments (4pKv), with novel modules (4pKv, 7NBx). We are also thankful for their judgment for well-written and easy-to-follow (HC91), and clear and detailed description (7NBx).

Following the insightful comments given by the reviewers to improve our work, we have made improvements to the revised version of our paper. We summarize the updates below.

- We have added Figure 9 in Appendix D.3 (simplified diagrams) to provide an intuitive visualization of our generation procedure.
- We have added more qualitative comparisons with style transfer methods, pretrained I2I model based on semantic masks in Figure 16 and Figure 17 in Appendix E.3.
- We have added qualitative results and analysis of an experiment on training FSeSim with cubemap projection in Appendix E.3.
- We have added more details in Experiments section and Appendix.

---

### Meta-Review · Area_Chair_ZiU2 · 2023-12-13

**Metareview:**

This paper proposes a panoramic I2I task by translating panoramic images to pinhole images as the target domain. A new method for non-paired panoramic image to pinhole image conversion is proposed, which solves the challenge of modifying original 360-degree panoramic images using easily obtainable pinhole images as the training target domain. The authors introduce a new model that uses spherical position embedding and sphere-based rotation enhancement to address the discontinuity problem at the edges of the panorama. The effectiveness of this method has been demonstrated through experiments. This paper has good motivation, is well-organized, and the experiments also prove the effectiveness of the proposed components. However, the experiments in this paper are not convincing enough, and the reviewers pointed out that there are some unfair comparisons in this paper.

**Justification For Why Not Higher Score:**

It seems that all reviewers have given negative reviews, strongly suggesting that the author should make more comprehensive comparisons of the experimental settings and provide a more in-depth explanation of the motivation behind the proposed method.

**Justification For Why Not Lower Score:**

N/A

---

### Decision · Program_Chairs · 2024-01-16

Reject